

# 1 Global modelling of soil carbonyl sulfide exchanges

Camille Abadie[1], Fabienne Maignan[1], Marine Remaud[1], Jérôme Ogée[2], J. Elliott Campbell[3],
Mary E. Whelan[4], Florian Kitz[5], Felix M. Spielmann[5], Georg Wohlfahrt[5], Richard Wehr[6], Wu
Sun[7], Nina Raoult[1], Ulli Seibt[8], Didier Hauglustaine[1], Sinikka T. Lennartz[9,10], Sauveur
Belviso[1], David Montagne[11] and Philippe Peylin[1].
[1]Laboratoire des Sciences du Climat et de l'Environnement, LSCE/IPSL, CEA-CNRS-UVSQ, Université Paris-
Saclay, Gif-sur-Yvette, France
[2]INRA, UMR 1391 ISPA, 33140 Villenave d'Ornon, France
[3]Sierra Nevada Research Institute, University of California, Merced, California 95343, USA
[4]Department of Environmental Sciences, Rutgers University, New Brunswick, NJ 08901, USA
[5]Department of Ecology, University of Innsbruck, Innsbruck, 6020, Austria
[6]Center for Atmospheric and Environmental Chemistry, Aerodyne Research, Inc., Billerica, Massachusetts, 01821,
USA.
[7]Department of Global Ecology, Carnegie Institution for Science, Stanford, CA 94305, USA
[8]Department of Atmospheric & Oceanic Sciences, University of California Los Angeles, California 90095, USA
[9]Institute of Chemistry and Biology of the Marine Environment, University of Oldenburg, 26129 Oldenburg,
Germany
[10]Department of Earth, Atmospheric and Planetary Sciences, Massachusetts Institute of Technology, Cambridge,
02139, MA, USA
[11]AgroParisTech, INRAE, Université Paris-Saclay, UMR ECOSYS, 78850 Thiverval-Grignon, France
*Correspondence to:* Camille Abadie (camille.abadie.research@gmail.com)
**Abstract.** Carbonyl sulfide (COS) is an atmospheric trace gas of interest for C cycle research because COS uptake
by continental vegetation is strongly related to terrestrial gross primary productivity (GPP), the largest and most
uncertain flux in atmospheric $CO_2$ budgets. However, to use atmospheric COS budgets as an additional tracer of
GPP, an accurate quantification of COS exchange by soils is also needed. At present, the atmospheric COS budget
is unbalanced globally, with total COS flux estimates from oxic and anoxic soils that vary between -409 and -104
GgS yr$^{-1}$. This uncertainty hampers the use of atmospheric COS concentrations to constrain GPP estimates through
atmospheric transport inversions. In this study we implemented a mechanistic soil COS model in the ORCHIDEE
land surface model to simulate COS fluxes in oxic and anoxic soils. Evaluation of the model against flux
measurements at 7 sites yields a mean root mean square deviation of 1.6 pmol m$^{-2}$ s$^{-1}$, instead of 2 pmol m$^{-2}$ s$^{-1}$
when using a previous empirical approach that links soil COS uptake to soil heterotrophic respiration. The new
model predicts that, globally and over the 2009-2016 period, oxic soils act as a net uptake of -126 GgS yr$^{-1}$, and
anoxic soils are a source of +96 GgS yr$^{-1}$, leading to a global net soil sink of only -30 GgS yr$^{-1}$, i.e., much smaller
than previous estimates. The small magnitude of the soil fluxes suggests that the error in the COS budget is
dominated by the much larger fluxes from plants, oceans, and industrial activities. The predicted spatial
distribution of soil COS fluxes, with large emissions in the tropics from oxic (up to 68.2 pmol COS m$^{-2}$ s$^{-1}$) and
anoxic (up to 36.8 pmol COS m$^{-2}$ s$^{-1}$) soils, marginally improves the latitudinal gradient of atmospheric COS
concentrations, after transport by the LMDZ atmospheric transport model. The impact of different soil COS flux
representations on the latitudinal gradient of the atmospheric COS concentrations is strongest in the northern
hemisphere. We also implemented spatio-temporal variations of near-ground atmospheric COS concentrations in
the modelling of biospheric COS fluxes, which helped reduce the imbalance of the atmospheric COS budget by
lowering COS uptake by soils and vegetation globally (-10% for soil, and -8% for vegetation with a revised mean
estimate of -576 GgS y$^{-1}$ over 2009-2016). Sensitivity analyses highlighted the different parameters to which each



soil COS flux model is the most responsive, selected in a parameter optimization framework. Having both
vegetation and soil COS fluxes modelled within ORCHIDEE opens the way for using observed ecosystem COS
fluxes and larger scale atmospheric COS mixing ratios to improve the simulated GPP, through data assimilation
techniques.

## 1   Introduction

Carbonyl sulfide (COS) has been proposed as a tracer for constraining the simulated Gross Primary Productivity
(GPP) in Land Surface Models (LSMs) (Launois et al., 2015; Remaud et al., 2021; Campbell et al., 2008). COS is
an atmospheric trace gas that is scavenged by plants at the leaf level through stomatal uptake and irreversibly
hydrolyzed in a reaction catalyzed by the enzyme Carbonic Anhydrase (CA) (Protoschill-Krebs et al., 1996). This
enzyme also interacts with $CO_2$ inside leaves. COS and $CO_2$ follow a similar pathway from the atmosphere to the
leaf interior. However, while $CO_2$ is also released during respiration, plants generally do not emit COS (Montzka
et al., 2007; Sandoval-Soto et al., 2005; Wohlfahrt et al., 2012). To infer GPP at the regional scale using COS
observations, modelers can use measurements of ecosystem COS fluxes directly, or measurements of atmospheric
COS concentrations combined with an atmospheric transport inversion model, provided all COS flux components
are taken into account. In both cases, net soil COS flux estimates are needed, as well as a functional relationship
between GPP and COS uptake by foliage.
One important limitation for using COS as a tracer for GPP is the uncertainty that remains on the COS budget
components. Several atmospheric transport inversion studies have suggested that a COS source located over the
tropical oceans and estimated between 700 and 1100 GgS yr$^{-1}$ was missing to close the contemporary COS budget
(Berry et al., 2013; Glatthor et al., 2015; Kuai et al., 2015). This hypothesis of a strong oceanic source has not
been substantiated by in situ COS and $CS_2$ measurements in sea waters (Lennartz et al., 2017, 2020, 2021), except
by Davidson et al. (2021) that invoke an oceanic source of $600 \pm 400$ GgS yr$^{-1}$ based on direct measurements of
sulfur isotopes. Clearly, an accurate characterization of all flux components of the atmospheric COS budget is still
needed. In particular, the contribution of soils to the COS budget is poorly constrained and improved estimates of
their contribution may therefore provide clues to the attribution of the missing source.
A distinction is usually made between oxic soils that mainly absorb COS, and anoxic soils that emit COS (Whelan
et al., 2018). Regarding COS uptake, COS diffuses into the soil, where it is hydrolyzed by CA contained in soil
microorganisms such as fungi and bacteria (Smith et al., 1999). It is to be noted that COS can also be consumed
by other enzymes, like nitrogenase, CO dehydrogenase, or $CS_2$ hydrolase (Smith and Ferry, 2000; Masaki et al.,
2021), but these enzymes are less ubiquitous than CA. The rate of uptake varies with soil type, temperature, and
soil moisture (Kesselmeier et al., 1999; VanDiest et al. 2007; Whelan et al., 2016). With high temperature or
radiation, soils were also found to emit COS through thermal or photo degradation processes (Kitz et al., 2017,
2020; Whelan and Rhew, 2015; Whelan et al., 2016, 2018). Although such COS emissions can be large in some
conditions, they are usually neglected in current figures of the atmospheric COS budget.
Using the empirical relationship between soil COS uptake and soil respiration by Yi et al. (2007), Berry et al.
(2013) provided new global estimates of COS uptake by oxic soils. Launois et al. (2015) proposed another
empirical model, linking oxic soil COS uptake to $H_2$ deposition based on the correlation between these two
processes observed at Gif-sur-Yvette (Belviso et al., 2013). Models with a physical representation of the involved
processes are also available. Sun et al. (2015) proposed such a mechanistic model including COS diffusion and



reactions within a layered soil. Ogée et al. (2016) also developed a mechanistic model including both COS uptake
and production, with steady-state analytical solutions in homogeneous soils. When including such models in an
LSM, the challenge is to spatialize them, which requires new variables or parameters not readily available at the
global scale but inferred form field or lab experiments.
In this study, our goal is to provide and evaluate new global estimates of net soil COS exchange. To this end:
i.  We implemented an empirical-based and a mechanistic-based soil COS model in the ORCHIDEE
LSM;
ii.  We evaluated the soil COS models at seven sites against in situ flux measurements;
iii.  We estimated soil contributions to the COS budget at the global scale;
iv.  We transported all COS sources and sinks using an atmospheric model and evaluated the
concentrations against measurements of the National Oceanic and Atmospheric Administration
(NOAA) air sampling network.
## 2  Methods
### 2.1  Description of the models
#### 2.1.1  The ORCHIDEE Land Surface Model
The ORCHIDEE Land Surface Model is developed at the Institut Pierre Simon Laplace (IPSL). The model version
used here is the one involved in the 6[th] Coupled Model Inter-comparison Project (CMIP6) (Boucher et al., 2020;
Cheruy et al., 2020). ORCHIDEE computes the carbon, water and energy balances over land surfaces. It can be
run at the site level or at the global scale. Fast processes such as soil hydrology, photosynthesis and respiration are
computed at a half-hourly time step. Other processes such as carbon allocation, leaf phenology and soil carbon
turnover are evaluated at a daily time step. Plant species are classified into 14 Plant Functional Types (PFTs),
according to their structure (trees, grasslands, croplands), bioclimatic range (boreal, temperate, tropical), leaf
phenology (broadleaf versus evergreen) and photosynthetic pathway ($C_3$ versus $C_4$). The vegetation distribution in
each grid cell is prescribed using yearly-varying PFT maps, derived from the ESA Climate Change Initiative (CCI)
land cover products (Poulter et al., 2015).
Soil parameters such as soil porosity, wilting point, and field capacity are derived from a global map of soil textures
based on the FAO/USDA (Food and Agriculture Organization of the United Nations/United States Department of
Agriculture) texture classification with 12 texture classes (Reynolds et al., 2000). The different textures for the
USDA classification are presented in Table S1 in the supporting information. To better represent the observed soil
conditions at the different sites, we substituted the soil textures initially assigned in ORCHIDEE from the USDA
texture global map with the observed soil textures corresponding to the USDA texture classes (Table S2). In a
previous study of vegetation COS fluxes in ORCHIDEE, Maignan et al. (2021) used the global soil map based on
the Zobler texture classification (Zobler, 1986), which is reduced to 3 different textures in ORCHIDEE. However,
the USDA soil classification gives a finer description of the different soil textures than the Zobler soil
classification, considering 12 soil textures instead of 3. The move from the coarse Zobler classes to the finer USDA
classes is found to be more important to the mechanistic model. Since the USDA texture classes are more accurate
with its finer discretization of soil textures, in the rest of this study, we only illustrate the results based on the
USDA texture classification.






For site level simulations, the ORCHIDEE LSM was forced by local micro-meteorological measurements obtained
from the FLUXNET network at the FLUXNET sites following the Creative Commons (CC-BY 4.0) license
(Pastorello et al., 2020), and at the remaining sites by other local meteorological measurements performed together
with the COS fluxes measurements when available, eventually gap-filled using the 0.25°x0.25°, hourly reanalysis
from the fifth generation of meteorological analyses of the European Centre for Medium-Range Weather Forecasts
(ECMWF) (ERA5) (Hersbach et al., 2020). Global simulations were forced by the 0.5° and 6-hourly CRUJRA
reanalysis (Friedlingstein et al., 2020). Near-surface COS concentrations (noted $C_a$ below) were prescribed using
monthly-mean atmospheric COS concentrations at the first vertical level of the LMDZ atmospheric transport
model (GCM, see description below in Sect. 2.1.3), forced with optimized COS surfaces fluxes. The latter have
been inferred by atmospheric inverse modelling from the COS surface measurements of the NOAA network
(Remaud et al., 2021). Simulations with constant atmospheric COS concentrations at a mean global value of 500
ppt were also run, to evaluate the impact of spatio-temporal variations of near-surface COS concentrations versus
a constant value. Near-surface $CO_2$ concentrations were estimated using global yearly-mean values provided by
the TRENDY project (Sitch et al., 2015).
**2.1.2   COS soil models**
**The empirical soil COS flux model**
We implemented in the ORCHIDEE LSM the soil COS flux model from Berry et al. (2013), which assumes that
COS uptake is proportional to $CO_2$ production by soil respiration, following Yi et al. (2007). Although Yi et al.
(2007) reported a relationship between soil COS uptake and total soil respiration, including root respiration, Berry
et al. (2013) assumed that COS flux was proportional to soil heterotrophic respiration only. The rationale behind
this assumption is that soil CA concentration is related to soil organic matter content, and thus ecosystem
productivity (Berry et al., 2013). As heterotrophic respiration is also linked to productivity, Berry et al. (2013)
considered soil COS uptake to be proportional to soil heterotrophic respiration. However, soil respiration alone
did not correlate well in incubation studies (Whelan et al., 2016). As the proportionality between COS fluxes and
soil respiration has only been demonstrated for the total (heterotrophic and autotrophic) soil respiration (Yi et al.
2007), we used in this study total soil respiration as a scaling factor for soil COS uptake. This model will be
referred to as the empirical model.

The influence of soil temperature and moisture are included in the calculation of soil respiration. Thus, we
computed soil COS flux $F_{soil,empirical}$ (pmol COS m$^{-2}$ s$^{-1}$) as follows,
$$F_{soil,empirical} = - k_{soil} * Resp_{tot} \qquad\qquad (1)$$
where $Resp_{tot}$ is total soil respiration (µmol $CO_2$ m$^{-2}$ s$^{-1}$) and $k_{soil}$ is a constant equal to 1.2 pmol COS µmol$^{-1}$
$CO_2$ that converts $CO_2$ production from respiration to COS uptake. The value of 1.2 pmol COS µmol$^{-1}$ was
estimated from field chamber measurements in a pine and broadleaf mixed forest (Dinghushan Biosphere Reserve,
south China) from Yi et al. (2007). In ORCHIDEE, we calculated the total soil respiration as the sum of soil
heterotrophic respiration within the soil column, including that of the litter, and root autotrophic respiration.





**The mechanistic soil COS flux model**
The mechanistic COS soil model of Ogée et al. (2016) describes both soil COS uptake and production. This model
includes COS diffusion in the soil matrix, COS dissolution and hydrolysis in the water-filled pore space and COS
production under low redox conditions. COS advection is neglected as the advective flux becomes negligible for
time scales longer than 1 h (Ogée et al., 2016). The soil is assumed to be horizontally homogeneous so that the
soil COS concentration $C$ (mol m$^{-3}$) is only a function of time $t$ (s) and soil depth $z$ (m). The mass balance equation
for COS can then be written as (Ogée et al., 2016),
$$\frac{\partial \varepsilon_{tot} C}{\partial t} = -\frac{\partial F_{diff}}{\partial z} + P - S \tag{2}$$

with $\varepsilon_{tot}$ the soil total porosity (m$^3$ air m$^{-3}$ soil), $F_{diff}$ the diffusional flux of COS (mol m$^{-2}$ s$^{-1}$), $S$ the COS
consumption rate (mol m$^{-3}$ s$^{-1}$) and $P$ the COS production rate under low redox conditions (mol m$^{-3}$ s$^{-1}$).
Under steady-state conditions and uniform soil temperature, moisture and porosity profiles, an analytical solution
of Eq. 2 can be found (Ogée et al., 2016). Although Eq. 2 could also be solved numerically using the soil
discretization in ORCHIDEE, we preferred to use the analytical solution, using the mean soil moisture and
temperature averaged over the first few soil layers (down to about 9 cm deep), weighted by the thickness of each
soil layer. Assuming fully mixed atmospheric conditions within and below the vegetated canopy, we also assumed
that the COS concentration at the soil surface $C(z = 0)$ is equal to the near-surface COS concentration $C_a$. With
these boundaries' conditions, the steady-state COS flux at the soil surface $F_{soil,mechanistic}$ (mol m$^{-2}$ s$^{-1}$) is (Ogée
et al., 2016),
$$F_{soil,mechanistic} = \sqrt{kB\theta D} \left( C_a - \frac{z_1^2 P}{D} \left( 1 - exp(-z_{max}/z_1) \right) \right) \tag{3}$$

with $k$ the first-order COS consumption rate constant within the soil (s$^{-1}$), $B$ the solubility of COS in water (m$^3$
water m$^{-3}$ air), $\theta$ the soil volumetric water content (m$^3$ water m$^{-3}$ soil), $D$ the total effective COS diffusivity (m$^2$
s$^{-1}$), $z_1 = \sqrt{D/kB\theta}$ (m) and $z_{max}$ the soil depth below which the COS production rate and the soil COS gradient
are assumed negligible (Ogée et al., 2016). In the following, $z_{max}$ is set at 0.09 m.
COS diffusion
The total effective COS diffusivity in soil, $D$, includes the effective diffusivity of gaseous COS $D_{eff,a}$ (m$^3$ air m$^{-1}$
soil s$^{-1}$) and dissolved COS $D_{eff,l}$ (m$^3$ water m$^{-1}$ soil s$^{-1}$) through the soil matrix,
$$D = D_{eff,a} + D_{eff,l} B \tag{4}$$

The solubility of COS in water $B$ is calculated using Henry's law constant $K_H$ (mol m$^{-3}$ Pa$^{-1}$),
$$B = K_H R T \tag{5}$$

with $R$ = 8,314 J mol$^{-1}$ K$^{-1}$ the ideal gas constant and $T$ the soil temperature (K) and (Wilhelm et al., 1977),
$$K_H = 0.00021 \, exp[24900/R(1/T - 1/298,15)] \tag{6}$$





The effective diffusivity of gaseous COS $D_{eff,a}$ is expressed as (Ogée et al., 2016),
$$D_{eff,a} = D_{0,a}\,\tau_a\,\varepsilon_a \qquad\qquad (7)$$
with $D_{0,a}$ the binary diffusivity of COS in the air (m$^2$ air s$^{-1}$), $\tau_a$ the air tortuosity factor representing the tortuosity
of the air-filled pores, and $\varepsilon_a$ is the air-filled porosity (m$^3$ air m$^{-3}$ soil). The binary diffusivity of COS in the air
$D_{0,a}$ is expressed following the Chapman-Enskog theory for ideal gases (Bird et al., 2002) and depends on
temperature and pressure,
$$D_{0,a}(T,p) = D_{0,a}(T_0,p_0) \left(\frac{T}{T_0}\right)^{1.5} \left(\frac{p}{p_0}\right) \qquad\qquad (8)$$
with $D_{0,a}(T_0,p_0) = D_{0,a}(25°C, 1\,atm) = 1.27 \times 10^{-5}$ m$^2$ s$^{-1}$ (Massman, 1998).
The expression of the air tortuosity factor $\tau_a$ depends on whether the soil is repacked or undisturbed. In
ORCHIDEE, repacked soils correspond to the agricultural soils represented by the C$_3$ and C$_4$ crops. Soils not
covered by crops are considered as undisturbed soils. The expression of $\tau_a$ for repacked soils $\tau_{a,r}$ is given by
Moldrup et al. (2003),
$$\tau_{a,r} = \varepsilon_a^{3/2}/\varphi \qquad\qquad (9)$$
with $\varphi$ the soil porosity (m$^3$ m$^{-3}$) that includes the air-filled and water-filled pores. Soil porosity is assumed constant
through the soil column in ORCHIDEE and is determined by the USDA texture global map. The air-filled porosity
$\varepsilon_a$ is calculated as $\varepsilon_a = \varphi - \theta$.
The expression of $\tau_a$ for undisturbed soils $\tau_{a,u}$ is given in Deepagoda et al. (2011). We chose this expression rather
than the expression proposed by Moldrup et al. (2003) for undisturbed soils because it appears to be more accurate
and does not require information on the pore-size distribution (Ogée et al., 2016),
$$\tau_{a,u} = [0.2(\varepsilon_a/\varphi)^2 + 0.004]/\varphi \qquad\qquad (10)$$
In a similar way to COS diffusion in the gas phase, the effective diffusivity of dissolved COS $D_{eff,l}$ is described
by Ogée et al. (2016),
$$D_{eff,l} = D_{0,l}\,\tau_l\,\theta \qquad\qquad (11)$$
with $D_{0,l}$ the binary diffusivity of COS in the free water (m$^2$ water s$^{-1}$) and $\tau_l$ the tortuosity factor for solute
diffusion. The binary diffusivity of COS in the free water $D_{0,l}$ is described using an empirical formulation proposed
by Zeebe (2011) for CO$_2$, which only depends on temperature,
$$D_{0,l}(T) = D_{0,l}(T_0) \left(\frac{T}{T_0} - 1\right)^2 \qquad\qquad (12)$$
with $T_0 = 216$K (Ogée et al., 2016) and $D_{0,l}(25°C) = 1.94 \times 10^{-9}$ m$^2$ s$^{-1}$ (Ulshöfer et al., 1996).
The expression of $\tau_l$ is the same for repacked and undisturbed soils. We used the expression given by Millington
and Quirk (1961) as a good compromise between simplicity and accuracy (Moldrup et al. 2003),



$\tau_l = \theta^{7/3}/\varphi^2$      (13)
COS consumption
COS can be destroyed by biotic and abiotic processes. The abiotic process corresponds to COS hydrolysis in soil
water at an uncatalyzed rate $k_{uncat}$ (s⁻¹), which depends on soil temperature $T$ (K) and $pH$ (Elliott et al., 1989),
$k_{uncat} = 2.15.10^{-5} exp(-10450(\frac{1}{T} - \frac{1}{298.15})) + 12.7.10^{-pK_w+pH} exp(-6040(\frac{1}{T} - \frac{1}{298.15}))$      (14)
with $pK_w$ the dissociation constant of water.
This uncatalyzed hydrolysis is quite low compared to the COS hydrolysis catalysed by soil microorganisms, which
is the main contribution of COS uptake by soils (Kesselmeier et al., 1999; Sauze et al., 2017; Meredith et al.,
2018). The enzymatic reaction catalysed by CA follows Michaelis-Menten kinetics. The turnover rate $k_{cat}$ (s⁻¹)
and the Michaelis-Menten constant $K_m$ (mol m⁻³) of this reaction depend on temperature. The temperature
dependence of the ratio $\frac{k_{cat}}{K_m}$ is expressed as (Ogée et al., 2016),
$x_{CA}(T) = \dfrac{exp(-\frac{\Delta H_a}{RT})}{1+exp(-\frac{\Delta H_d}{RT}+\frac{\Delta S_d}{R})}$      (15)
where $\Delta H_a$, $\Delta H_d$ and $\Delta S_d$ are thermodynamic parameters, such as $\Delta H_a$ = 40 kJ mol⁻¹, $\Delta H_d$ = 200 kJ mol⁻¹ and $\Delta S_d$
= 660 J mol⁻¹ K⁻¹.
The total COS consumption rate by soil $k$ (s⁻¹) is described with respect to the uncatalyzed rate at $T$ = 298.15 K
and $pH$ = 4.5 (Ogée et al., 2016),
$k = f_{CA} k_{uncat}(298.15, 4.5) \frac{x_{CA}(T)}{x_{CA}(298,15)}$      (16)
where $f_{CA}$ is the CA enhancement factor, which characterizes the soil microbial community that can consume
COS. The CA enhancement factor depends on soil CA concentration, temperature, and pH. Ogée et al. (2016)
reported that its values range between 21 600 and 336 000, with a median value at 66 000. We adapted the values
of $f_{CA}$ found in (Meredith et al., 2019) to have a CA enhancement factor that depends on ORCHIDEE biomes
(Appendix A, Table A1).
Oxic soil COS production
Abiotic oxic soil COS production has been observed at high soil temperature (Maseyk et al., 2014; Whelan and
Rhew, 2015; Kitz et al., 2017, 2020; Spielmann et al., 2019, 2020). However, photodegradation has also been
proposed as an abiotic production mechanism in oxic soils (Whelan and Rhew, 2015; Kitz et al., 2017, 2020).
Abiotic COS production is still not well understood but was assumed to originate from biotic precursors (Meredith
et al., 2018).
In Ogée et al. (2016), the production rate $P$ is described as independent of soil $pH$ but depends on soil temperature
and redox potential. This dependence on soil redox potential enables us to consider the transition between oxic
and anoxic soils. However, because little information is available on soil redox potential at the global scale, its
influence cannot yet be represented in a spatially and temporally dynamic way in a land surface model such as





ORCHIDEE. Thus, we decided to use the production rate described in Whelan et al. (2016) that only depends on
soil temperature and land use type,
$P_{oxic} = e^{\alpha + \beta T}$ (17)
where $P_{oxic}$ is expressed in pmol g$^{-1}$ min$^{-1}$, $T$ is soil temperature (°C) and $\alpha$ and $\beta$ are parameters determined by
Whelan et al. (2016) for each land use type using the least-squares fitting approach. We adapted the values of $\alpha$
and $\beta$ given for four land use types to ORCHIDEE biomes (Appendix A Table A2). Values of $\alpha$ and $\beta$ for deserts
could not be estimated by Whelan et al. (2016) because COS emission for this biome was not found to increase
with temperature. Figure 11 in Whelan et al. (2016) shows that COS emission from a desert soil is always near
zero for temperatures ranging from 10°C to 40°C. Moreover, COS emission from a desert soil is also found to be
near zero in Fig. 1 of Meredith et al. (2018). This could be explained by a lack of organic precursors to produce
COS (Whelan et al., 2016). Therefore, we considered that desert soils, which correspond to a specific non-
vegetated PFT in ORCHIDEE, do not emit COS. For other ORCHIDEE biomes, COS production was estimated
using $\alpha$ and $\beta$ for each PFT and the mean soil temperature over the top 9 cm. The unit of $P_{oxic}$ was converted from
pmol g$^{-1}$ min$^{-1}$ to mol m$^{-3}$ s$^{-1}$ (in equation 3) using soil bulk density information from the Harmonized World Soil
Database (HWSD; FAO/IIASA/ISRIC/ISSCAS/JRC, 2012).

Anoxic soil COS production
Several studies have shown direct COS emissions by anoxic soils (Devai and DeLaune, 1997; de Mello and Hines,
1994; Whelan et al., 2013; Yi et al., 2007). This has been linked to a strong activity of sulfate reduction
metabolisms in highly reduced environments such as wetlands (Aneja et al., 1981; Kanda et al., 1992; Whelan et
al., 2013; Yi et al., 2007). A previous approach developed by Launois et al. (2015) was based on the representation
of seasonal methane emissions by Wania et al. (2010) in the LPJ–WHyME model to represent anoxic soils in
ORCHIDEE. The mean values of soil COS emissions from Whelan et al. (2013) were used to attribute to each
grid point a value of soil COS emission. In this approach by Launois et al. (2015), salt marshes were not represented
despite their strong COS emissions found in Whelan et al. (2013). Emissions from rice paddies were also neglected.
Thus, COS emissions from anoxic soils peaked in summer over the high latitudes, following methane production.
Because of the scarce knowledge on anoxic soil COS exchange, here we propose another approach to represent
the contribution of anoxic soils, which could be compared to the previous approach developed by Launois et al.
(2015). To represent the distribution of anoxic soils we selected the regularly flooded wetlands from the map
developed by Tootchi et al. (2019), as represented in Fig. 1. The regularly flooded wetlands cover 9.7% of the
global land area, which is among the average values found in the literature ranging from 3% to 21% (Tootchi et
al., 2019). The pixels defined as anoxic soils are considered flooded through the entire year: the seasonal variations
of the flooding, as happening during the monsoon seasons, are consequently neglected.
The production rate for anoxic soils is based on the expression developed by Ogée et al. (2016),
$P_{anoxic} = P_{ref} z_{max} Q_{10}^{\frac{(T - T_{ref})}{10}}$ (18)



with $P_{ref}$ (mol m$^{-2}$ s$^{-2}$) the reference production term, $T_{ref}$ a reference soil temperature (K) and $Q_{10}$ the
multiplicative factor of the production rate for a 10 °C increase in soil temperature (unitless). As anoxic soil
production ranges from 10 to 300 pmol m$^2$ s$^{-1}$ for salt marshes and is usually below 10 pmol m$^{-2}$ s$^{-1}$ for freshwater
wetlands (Whelan et al., 2018), the reference production term was set to 10 pmol m$^{-2}$ s$^{-1}$.
All the variables and constants of the empirical and mechanistic models are presented in Appendix A Tables A3
and A4.

### 2.1.3    The atmospheric chemistry transport model LMDZ

To simulate the COS atmospheric distribution, we use an "offline" version of the Laboratoire de Météorologie
Dynamique General Circulation Model (GCM), LMDZ 6 (Hourdin et al., 2020), which has been used as the
atmospheric component in the IPSL Coupled Model for CMIP6. The LMDZ GCM has a spatial resolution
3.75°long.×1.9°lat. with 39 sigma-pressure layers extending from the surface to about 75 km, corresponding to a
vertical resolution of about 200-300 m in the planetary boundary layer, and a first level at 33 m above sea or
ground level. The model u and v wind components were nudged towards winds from ERA5 reanalysis with a
relaxation time of 2.5 hours to ensure realistic wind advection (Hourdin and Issartel, 2000; Hauglustaine et al.,
2004). The ECMWF fields are provided every 6 hours and interpolated onto the LMDZ grid. This version has
been shown to reasonably represent the transport of passive tracers (Remaud et al., 2018). The off-line model uses
pre-computed mass-fluxes provided by this full LMDZ GCM version and only solves the continuity equation for
the tracers, which significantly reduces the computation time. In the following, we refer to this offline version as
LMDZ. The model time step is 30 minutes, and the output concentrations are 3-hourly averages.
The atmospheric COS oxidation is computed from pre-calculated OH monthly concentration fields produced from
a simulation of the INCA (Interaction with Chemistry and Aerosols) model (Folberth et al., 2006; Hauglustaine et
al., 2004, 2014) coupled to LMDZ. The atmospheric OH oxidation of COS amounts to 100 GgS yr$^{-1}$ in the model.
Similarly, the COS photolysis rates are also pre-calculated with the INCA model, which uses the Troposphere
Ultraviolet and Visible (TUV) radiation model (Madronich et al., 2003) adapted for the stratosphere (Terrenoire
et al., in prep.). The temperature-dependent carbonyl sulfide absorption cross-sections from 186.1 nm to 296.3 nm
are taken from (Burkholder et al., 2019). The calculated photolysis rates are averaged over the period 2008-2018
and prescribed to LMDZ. Implemented in LMDZ, the COS photolysis in the stratosphere amounts to about 30
GgS yr$^{-1}$, which of the same order of magnitude as previous estimates: 21 GgS yr$^{-1}$ (71% of 30 GgS yr$^{-1}$) by Chin
and Davis (1995), between 11 GgS yr$^{-1}$ and 21 GgS yr$^{-1}$ by Kettle et al. (2002) and between 16 GgS yr$^{-1}$ and 40
GgS yr$^{-1}$ by Ma et al. (2021).

### 2.2    Observation data sets

### 2.2.1    Description of the sites

The description of the studied sites is given in Table 1.

### 2.2.2    Soil COS flux determination at selected sites

Soil COS flux chamber measurements were conducted in 2015 at AT-NEU, in 2016 at DK-SOR, ES-LMA and
ET-JA, and in 2017 at IT-CRO (abbreviations as in Table 1). The aboveground vegetation was removed one day
before the measurements if needed and the fluxes were derived from concentration measurements using a Quantum



Cascade Laser (see Kitz et al., 2020 and Spielmann et al., 2020, 2019). At AT-NEU, DK-SOR, ES-LMA and IT-
CRO, a Random Forest model was calibrated against the manual chamber measurements, and then used to simulate
half-hourly soil COS fluxes in Spielmann et al. (2019). We compared the ORCHIDEE half-hourly simulated fluxes
to half-hourly outputs of the Random Forest model. This enabled to study the diel cycle, and to compute daily
observations with no sampling bias for the study of the seasonal cycle. Soil COS fluxes for ET-JA were derived
by using the same training method than the one used in Spielmann et al. (2019).
At FI-HYY, soil COS fluxes were measured using two automated soil chambers in 2015. These chambers were
connected to a quantum cascade laser spectrometer to calculate soil COS fluxes from concentration measurements
(see Sun et al. (2018) for more information on the experimental setup).
At US-HA, soil COS fluxes were not directly measured but derived from eddy covariance COS and $CO_2$
measurements and soil chamber $CO_2$ measurements conducted in 2012 and 2013. A sub-canopy flux gradient
approach was used to partition canopy uptake from soil COS fluxes. For more information on this approach and
its limitations, see Wehr et al. (2017).
In the study of soil COS fluxes, the difficulty of performing soil COS flux measurements must be acknowledged,
as well as the differences between experimental setups and methods to retrieve soil COS fluxes. These limitations
are illustrated in the set of observations selected here. Aboveground vegetation had to be removed at some sites to
not measure the plant contribution in addition to soil COS fluxes (Sun et al., 2018; Spielmann et al., 2019; Kitz et
al., 2020). Vegetation removal prior to the measurements might lead to artefacts in the observations. Some
components of the measuring system can also emit COS. In this case, a blank system is needed to apply a post-
correction to the measured fluxes (Sun et al., 2018; Kitz et al., 2020). Litter was left in place at the measurement
sites.

### 2.2.3    COS concentrations at the NOAA/ESRL sites

The NOAA surface flask network provides long-term measurements of the COS mole fraction at 14 locations at
weekly to monthly frequencies from the year 2000 onwards. We use an extension of the data initially published in
Montzka et al. (2007). The data were collected as paired flasks analyzed using gas chromatography and mass
spectrometry. The stations located in the northern Hemisphere sample air masses coming from the entire northern
hemisphere domain above 30 degrees. Among them, the sites LEF, NWR, HFM, WIS have a mostly continental
footprints (Remaud et al., 2021) while the sites SPO, CGO, PSA sample mainly oceanic air masses of the southern
hemisphere (Montzka et al., 2007). The locations of these sites are depicted in Appendix B, Fig. B1.

## 2.3    Simulations

### 2.3.1    Spin-up phase

A "spin-up" phase was performed before each simulation, which enabled all carbon pools to stabilize and the net
biome production to oscillate around zero. Reaching the equilibrium state is accelerated in the ORCHIDEE LSM
thanks to a pseudo-analytical iterative estimation of the carbon pools, as described in Lardy et al. (2011). For site
simulations, the spin-up was performed by cycling the years available in the forcing files of each site, for a total
of about 340 years. For global simulations, the spin-up phase of 340 years was performed by cycling over 10 years
of meteorological forcing files in the absence of any disturbances.





**362**     **2.3.2**     **Transient phase**

**363**     Following the spin-up phase we ran a transient simulation of about 40 years that introduced disturbances such as

**364**     climate change, land use change and increasing $CO_2$ atmospheric concentrations.

**365**     This transient phase was performed by cycling over the available years for site simulations. For global simulations,

**366**     the transient phase was run where we introduced disturbances from 1860 to 1900. After this transient phase, COS

**367**     fluxes were simulated from 1901 to 2019.

**368**     **2.3.3**     **Atmospheric simulations: sampling and data processing**

**369**     We ran the LMDZ6 version of the atmospheric transport model described above for the years 2009 to 2016. We

**370**     started from a uniform initial condition and we remove the first year as it is considered to be part of the spin-up

**371**     period. The prescribed COS fluxes used as model inputs are presented in Table 2. The fluxes are given as a lower

**372**     boundary condition, called the surface, of the atmospheric transport model (LMDZ), which then simulates the

**373**     transport of COS by large-scale advection and sub-grid scale processes such as convection and boundary layer

**374**     turbulence. In this study, we only evaluate the sensitivity of the latitudinal gradient and seasonal cycle of COS

**375**     concentrations to the soil COS fluxes. The horizontal gradient aims at validating the latitudinal repartition of the

**376**     surface fluxes, while the seasonal cycle partly reflects the seasonal exchange with the terrestrial sink, which peaks

**377**     in spring/summer. This study does not aim at reproducing the mean value as the top-down COS budget is currently

**378**     unbalanced, with a source component missing (Whelan et al., 2018; Remaud et al., 2021, and see Table 5).

**379**     For each COS observation, the 3D simulated concentration fields were sampled at the nearest grid point to the

**380**     station and at the closest hour of the measurements. For each station, the curve fitting procedure developed by the

**381**     NOAA Climate Monitoring and Diagnostic Laboratory (NOAA/CMDL) (Thoning et al., 1989) was applied to

**382**     modelled and observed COS time series to extract a smooth detrended seasonal cycle. We first fitted a function

**383**     including a first-order polynomial term for the growth rate and two harmonic terms for seasonal variations. The

**384**     residuals (raw time series minus the smooth curve) were fitted using a lowpass filter with either 80 or 667 d as

**385**     short-term and long-term cut-off values. The detrended seasonal cycle is defined as the smooth curve (full function

**386**     plus short-term residuals) minus the trend curve (polynomial plus long-term residuals). Regarding vegetation COS

**387**     fluxes (Maignan et al., 2021), we added the possibility to use spatially and temporally varying atmospheric COS

**388**     concentrations, as for soil.

**389**     **2.4**     **Numerical methods for model evaluation and parameter optimisation**

**390**     **2.4.1**     **Statistical scores**

**391**     We evaluated modelled soil COS fluxes against field measurements using the Root Mean Square Deviation

**392**     (RMSD):

**393**     $$RMSD = \sqrt{\frac{\sum_{n=1}^{N}\left(F_{COS}^{Obs}(n) - F_{COS}^{Mod}(n)\right)^2}{N}}$$     (19)

**394**     where $N$ is the number of considered observations, $F_{COS}^{Obs}(n)$ is the $n$th observed COS flux and $F_{COS}^{Mod}(n)$ is the $n$th

**395**     modelled COS flux, and the relative RMSD (rRMSD):



$rRMSD = \dfrac{RMSD}{\frac{\sum_{n=1}^{N} F_{COS}^{Obs}(n)}{N}}$                                                      (20)
which is the RMSD divided by the mean value of observations.
Simulated atmospheric COS concentrations were evaluated by computing the normalized standard deviations
(NSDs), which is the standard deviation of the simulated concentrations divided by the mean of the observed
concentrations, and the Pearson correlation coefficients (r) between simulated and observed COS concentrations.
The closer NSD and r values are to 1, the better the model accuracy is.

### 402     2.4.2     Data assimilation

One of the main difficulties with the implementation of a model is to define the parameter values that lead to the
most accurate representation of the processes in ORCHIDEE. Calibrating the model parameters is of interest as
Ogée et al. (2016) indicate that some of the model parameters such as $f_{CA}$ and the production term parameters have
to be constrained by observations. Moreover, the default values for the soil COS model parameters used in this
study (Appendix A Tables A1 and A2) are determined by laboratory experiments (Ogée et al., 2016; Whelan et
al., 2016), that is why it is interesting to study how the values obtained by calibration against field observations
differ from these default values. Data assimilation (DA) aims at producing an optimal estimate by combining
observations and model outputs. In this study, we used data assimilation to find the model parameter values that
improve the fit between simulated and observed soil COS fluxes from the empirical and the mechanistic models.
We used the ORCHIDEE DA System (ORCHIDAS), which is based on a Bayesian framework. ORCHIDAS has
been described in detail in previous studies (Bastrikov et al., 2018; Kuppel et al., 2014; MacBean et al., 2018;
Peylin et al., 2016; Raoult et al., 2021), so below we only briefly present the method. Assuming that the
observations and model outputs follow a Gaussian distribution, we aim at minimizing the following cost function
$J(x)$ by optimizing the model parameters (Tarantola, 2005),
$J(x) = \frac{1}{2} [(M(x) - y)^T \cdot E^{-1} \cdot (M(x) - y) + (x + x^b)^T \cdot B^{-1} \cdot (x + x^b)]$                    (21)
with $x$ the vector of parameters to optimize and $y$ the observations. The first part of the cost function measures the
mismatch between the observations and the model, and the second part represents the mismatch between the prior
parameter values $x^b$ and the considered set of parameters $x$. Both terms of the cost function are weighted by the
prior covariance matrices for the observation errors $E^{-1}$ and parameter errors $B^{-1}$. The minimization of the cost
function follows the genetic algorithm (GA) method, which is derived from the principles of genetics and natural
selection (Goldberg, 1989; Haupt and Haupt, 2004) and is described for ORCHIDAS in Bastrikov et al. (2018).
For each soil COS model, we selected the 8 most important parameters to which soil COS fluxes are sensitive
following sensitivity analyses (Sect. 2.4.3). The observation sites selected for sensitivity analyses and DA are the
ones with the largest number of observations for model parameter calibration, which are FI-HYY and US-HA.

### 427     2.4.3     Sensitivity analyses

We conducted sensitivity analyses at two contrasting sites (FI-HYY and US-HA) to determine which model
parameters have the most influence on the simulated soil COS fluxes from the empirical and the mechanistic
models. Sensitivity analyses can help to identify the key parameters before aiming at calibrating these parameters.



Indeed, focusing on the key model parameters for calibration limits both the computational cost of optimization
that increases with the number of parameters and the risk of overfitting.
The Morris method (Morris, 1991; Campolongo et al., 2007) was used for the sensitivity analysis as it is relatively
time-efficient and enables ranking the parameters by importance. This qualitative method requires only a small
number of simulations, $(p+1)n$, with $p$ the number of parameters and $n$ the number of random trajectories generated
(here, $n$=10).
We selected a set of parameters for the Morris sensitivity analyses based on previous sensitivity analyses conducted
on soil parameters in ORCHIDEE (Dantec-Nédélec et al., 2017; Raoult et al., 2021; Mahmud et al., 2021). A
distinction is made between the soil COS model parameters called first-order parameters ($f_{CA}$, $\alpha$ and $\beta$ for the
mechanistic model and $k_{soil}$ for the empirical model), and parameters called second-order parameters related to
soil hydrology, carbon uptake and allocation, phenology, conductance, or photosynthesis (18 parameters, see
Tables S3 and S4). The range of variation of the second-order parameters are described in previous studies using
ORCHIDEE (Dantec-Nédélec et al., 2017; Raoult et al., 2021; Mahmud et al., 2021). For the first-order
parameters, the range of variation is described in Yi et al. (2007) for $k_{soil}$ (±1.08 pmol COS µmol$^{-1}$ CO$_2$) and in
Table 1 in Meredith et al. (2019) for $f_{CA}$. The ranges of variation for $\alpha$ and $\beta$ parameters are not directly given in
the literature and were calculated based on information from the production parameters defined in Meredith et al.
(2018) (Text S1 and Table S5).
**3    Results**
**3.1    Site scale COS fluxes**
**3.1.1    Soil COS flux seasonal cycles**
Figure 2 shows the seasonal cycles of soil COS fluxes at the different sites where measurements were conducted.
The empirical model mainly differs from the mechanistic model with a stronger seasonal amplitude of soil COS
fluxes (34% higher), except at the sites where a net COS production is found with the mechanistic model in summer
(ES-LMA and IT-CRO). At all sites, the empirical model shows that the simulated uptake increases in spring
reaching a maximum in summer, and decreases in autumn with a minimal uptake during winter. The strong COS
uptake in summer from the empirical model can be explained by the proportionality of soil COS uptake to
simulated soil respiration, which increases with the high temperatures in summer. In contrast, the mechanistic
model depicts almost no seasonality at all the sites where no net COS production is found over the year. As the
mechanistic model represents both soil COS uptake and production, the increase in COS production due to higher
temperature in summer compensates part of the COS uptake (Appendix C Figure C1). While the uptake from the
empirical model is often higher than the one computed with the mechanistic model in summer, soil COS uptake
in winter is stronger with the mechanistic representation.
The scarcity of field measurements at AT-NEU, ES-LMA, IT-CRO, DK-SOR and ET-JA does not allow an
evaluation of the simulated seasonality of COS fluxes. However, at US-HA, the absence of seasonality from May
to October in the observations is also found in the mechanistic model, while a maximum net soil COS uptake is
reached with the empirical model.
We found that the mechanistic model is in better agreement with the observations for 4 (IT-CRO, ET-JA, FI-HYY,
US-HA) out of the 7 sites (Table 3), with a mean of 1.58 pmol m$^{-2}$ s$^{-1}$ and 2.03 m$^{-2}$ s$^{-1}$ for the mechanistic and
empirical model, respectively. However, the mechanistic model struggles to reproduce soil COS fluxes at AT-





NEU and ES-LMA, with an overestimation of soil COS uptake or an underestimation of soil COS production at
AT-NEU and a delay in the simulated net COS production at ES-LMA. We might suspect that the removal of
vegetation at these sites prior to the measurements could have artificially enhanced COS production in the
observations. The mechanistic model is able to represent a net COS production at IT-CRO but overestimates it.
This might highlight the importance of adapting the production parameters ($\alpha$ and $\beta$) in this model to adequately
represent a net COS production. As expected, the empirical model is unable to correctly simulate the direction of
the observed positive soil COS exchange rates at ES-LMA and IT-CRO.

### 3.1.2    Soil COS flux diel cycles

Figure 3 shows the comparison between the simulated and observed mean diel cycles over a month. The
observations show a minimum net soil COS uptake or a maximum net soil COS production reached between 11
am and 1 pm at AT-NEU, ES-LMA, IT-CRO and DK-SOR. A minimum net soil COS uptake is also observed at
US-HA but in the afternoon. At AT-NEU and ES-LMA, neither model is able to represent the observed diel cycle.
At IT-CRO, DK-SOR and US-HA, the diel cycles simulated by the mechanistic model show patterns similar to
the observations with a peak in the middle of the day, but with an overestimation of the net soil COS production
and a delay in the peak at IT-CRO, and an overestimation of the net soil COS uptake at DK-SOR. The mechanistic
model reproduces the absence of a diel cycle observed at FI-HYY. Small diel variations are observed at ET-JA,
which are also captured by the mechanistic model but with an underestimation of the net soil COS uptake. As the
mechanistic model includes PFT-specific parameters ($f_{CA}$, $\alpha$, $\beta$), we can think that these parameters would need
to be calibrated to improve the model performance at the site-scale. The empirical model shows a maximum soil
COS uptake around 3 pm at ET-JA, FI-HYY, US-HA and IT-CRO, which is not found in the observations at FI-
HYY and is in contradiction with the observed diel variations at IT-CRO and ES-LMA. Considering all sites, the
mechanistic model leads to a smaller error between the simulations and the observations, with a mean RMSD of
1.38 pmol m$^2$ s$^{-1}$ against 1.87 pmol m$^2$ s$^{-1}$ for the empirical model (Table 4).

### 3.1.3    Dependency on environmental variables

Figure 4 represents simulated net soil COS fluxes versus soil temperature and soil water content at the different
sites. At the sites where only a net soil COS uptake is simulated by the mechanistic model (all sites except IT-
CRO and ES-LMA), soil COS uptake globally decreases with increasing soil water content, which appears to be
the main driver of soil COS fluxes. This behaviour can be explained by a decrease in COS diffusivity through the
soil matrix with increasing soil moisture, reducing soil COS availability for microorganism consumption.
Furthermore, an optimum soil water content for net soil COS uptake is found between 10% and 15%. This optimum
soil moisture is also represented in Ogée et al. (2016) and was described in several field studies to be around 12%
(Kesselmeier et al., 1999; Liu et al., 2010; van Diest and Kesselmeier, 2008). The optimum soil water content for
soil COS uptake is related to a site-specific temperature optimum, which is found between 13°C and 15°C at US-
HA for example. Similarly, a temperature optimum was described in Ogée et al. (2016) and in empirical studies
with an optimum value that also depends on the studied site (Kesselmeier et al., 1999; Liu et al., 2010; van Diest
and Kesselmeier, 2008). At IT-CRO and ES-LMA where a strong net soil COS production is simulated by the
mechanistic model, the main driver of soil COS fluxes becomes soil temperature. At these sites, the net soil COS



production increases with soil temperature, due to the exponential response of soil COS production term to soil
temperature.
Contrary to the mechanistic model, soil COS uptake computed with the empirical model is mainly driven by soil
temperature, with a soil COS uptake that increases with increasing soil temperature. This response of the empirical
model to soil temperature is due to its relation to soil respiration, which is enhanced by strong soil temperature.
However, low soil moisture values were found to limit soil COS uptake for the empirical model, as seen at ES-
LMA for a soil water content below 8%.

### 3.1.4     Sensitivity analyses of soil COS fluxes to parameterization

Sensitivity analyses including a set of parameters (19 for the empirical model and 21 for the mechanistic model)
were performed to evaluate the sensitivity of soil COS fluxes to each of the selected parameter. The Morris scores
were normalised by highest values to help rank the parameters by their relative influence on soil COS fluxes, a
score of 1 represents the most important parameter and 0 represents the parameters that have no influence on soil
COS fluxes. For reasons of clarity, in the following we present the results only for the parameters that were found
to have an impact on soil COS fluxes (Morris scores not equal to 0).

Figure 5 shows the results of the Morris sensitivity experiments highlighting the key parameters influencing soil
COS fluxes from the empirical and the mechanistic models at FI-HYY and US-HA. For the empirical model at
both sites, the first order parameter ($k_{soil}$) is the most important parameter in the computation of soil COS fluxes,
as it directly scales soil respiration to soil COS fluxes. The following parameters to which soil COS fluxes are the
most sensitive are the scalar on the active soil C pool content (soilC) and the temperature-dependency factor for
heterotrophic respiration (soil_Q10). Indeed, the soilC parameter determines the soil carbon active pool content,
which can be consumed by soil microorganisms during respiration, therefore impacting soil COS fluxes from the
empirical model. soil_Q10 impacts soil COS fluxes at both sites as it determines the response of soil heterotrophic
respiration to temperature, which is included in the proportionality of soil COS fluxes to the total soil respiration
in the empirical model. Similarly, one of the second order parameters, the minimum soil wetness to limit the
heterotrophic respiration (min_SWC_resp), has an impact on soil COS fluxes from the empirical model only. The
importance of min_SWC_resp for soil COS fluxes is found at US-HA but not at FI-HYY. This can be explained
by the difference in soil moisture between the two sites, with an annual mean of 16.2% at US-HA and reaching a
minimum of only 8.8%, against an annual mean of 17.5% with a minimum of 12.4% at FI-HYY.
Contrary to the empirical model, soil COS fluxes computed with the mechanistic model are more sensitive to two
second-order parameters, the Van Genuchten water retention curve coefficient n (n) and the saturated volumetric
water content (θSAT). These two second-order parameters are strongly linked to soil hydrology and determine the
soil water content, which affects COS diffusion through the soil matrix and its uptake. The Van Genuchten
coefficients occur in the relationships linking hydraulic conductivity and diffusivity to soil water content (van
Genuchten, 1980). At both sites, the strong impact of the Van Genuchten water retention curve coefficient n on
soil COS fluxes simulated with the mechanistic model highlights the critical importance of soil architecture. Thus,
soil COS fluxes computed with the mechanistic model are expected to strongly vary according to the different soil
types. Then, the first-order parameters ($f_{CA}$, $\alpha$ and $\beta$) also influence soil COS fluxes from the mechanistic model.
However, the uptake parameter ($f_{CA}$ of PFT 15, boreal $C_3$ grass) has the most influence on soil COS fluxes at FI-





HYY, while it is the production-related parameter ($\alpha$ of PFT 6, temperate broadleaved summergreen forest) that
has the largest impact at US-HA. The stronger influence of the production parameter involved in the temperature
response at US-HA might be explained by the difference of temperature between the two sites, which ranges from
-10°C to 25°C at US-HA with an annual mean of 7.5°C in 2013, while only ranging from -5°C to 15°C with an
annual mean of 4.3°C at FI-HYY in 2015. Similar to the difference in the main driver of soil COS fluxes found in
Fig. 4, the most important first-order parameters to which soil COS fluxes are sensitive seem to differ between
uptake and production parameters depending on the site conditions. It is to be noted that at US-HA, the most
important production parameters are the ones of the dominant PFT at this site (PFT 6), which also correspond to
a stronger response of the production term to temperature than for PFT 10 (temperate $C_3$ grass). However, at FI-
HYY the most influential uptake parameter is for PFT 15 that only represents 20% of the PFTs at this site while
PFT 7 (boreal needleleaf evergreen forest) is the dominant PFT. This can be explained by the range of variation
that is assigned to $f_{CA}$ of PFT 7 by Meredith at al. (2019), which is larger than the one of $f_{CA}$ for PFT 15 (9000
against 3100).
Finally, a set of parameters related to photosynthesis, conductance, phenology, hydrology, and carbon uptake has
an impact on soil COS fluxes computed with both the empirical and the mechanistic models at the two sites. The
specific leaf area (SLA), maximum rate of Rubisco activity-limited carboxylation at 25°C (Vcmax25), residual
stomatal conductance (g0) and minimum photosynthesis temperature (Tmin) have an impact on soil COS fluxes
as they also indirectly affect soil moisture through their influence on transpiration and stomatal opening. The
second-order parameters related to soil hydrology (a, Ks, Zroot, θWP, θFC, θR, θTransp_max) impact the soil
water availability, which affects soil respiration for the empirical model and soil COS diffusion and uptake in the
mechanistic model. For example, the parameter for root profile (Zroot) determines the density and depth of the
roots, and therefore how much water can be taken up by roots.
**3.1.5    Soil COS flux optimization**
Figure 6 presents soil COS fluxes before and after optimization of the model parameters to better fit the
observations at FI-HYY and US-HA. For the mechanistic model, the optimization at the two sites mainly changes
the mean value of soil COS fluxes, by reducing the net uptake at US-HA and increasing it at FI-HYY. Similar to
the mechanistic model optimization, the posterior soil COS uptake computed with the empirical model is enhanced
at FI-HYY and reduced at US-HA. However, at US-HA, the increase in soil COS uptake is only found between
April and October, while the winter soil COS fluxes are not impacted by the optimization. Using the optimized
parameterization improves the RMSD by 7% and 5% at US-HA and by 23% and 25% at FI-HYY for the
mechanistic and the empirical model, respectively. While it leads to similar posterior RMSD values between the
two models at US-HA, the optimization of the mechanistic model gives a lower RMSD than the empirical model
at FI-HYY, with 0.54 pmol m$^{-2}$ s$^{-1}$ against 0.95 pmol m$^{-2}$ s$^{-1}$.
At FI-HYY, the difference between prior and posterior soil COS fluxes from the empirical model seems to mainly
come from the change in soil_Q10 value (Appendix E, Figure E1). soil_Q10 value drops from 0.83 to 0.53, which
corresponds to a prior Q10 value of 2.29 versus a posterior value of 1.70, decreasing the heterotrophic respiration
response to soil temperature. Soil COS fluxes computed with the empirical model were found to be strongly
sensitive to soil_Q10 (Figure 5). The posterior value of this parameter has nearly attained the lower bound of its
variation range. Since the range of variation represents the realistic values this parameter can take, we need to be



careful about the fact that this parameter is trying to take values close to, or potentially beyond, these meaningful
values. Furthermore, the optimization deviates the Q10 value at FI-HYY from the ones calculated in the
observations over the measurement period (3.0 for soil chamber 1 and 2.5 for soil chamber 2). We could assume
that $k_{soil}$ should be defined as temperature-dependent for linking soil COS flux to soil respiration (Berkelhammer
et al., 2014; Sun et al., 2018), instead of being considered as a constant. Thus, the optimization of the empirical
model could in fact be aliasing the error of $k_{soil}$ onto soil_Q10 because of the impossibility to account for the
temperature-dependence of soil COS to $CO_2$ uptake ratio (Sun et al., 2018). At US-HA, the optimization also leads
to a decrease of soil_Q10 but to a lesser extent, the parameter remaining comfortably within its range of variation.
For the mechanistic model, the optimization reduces the enhancement factor value ($f_{CA}$) for PFT 10 at US-HA and
increases the value of the production parameter $\alpha$ for the dominant PFT (PFT 6). This enhances the reduction in
net soil COS uptake, which was slightly overestimated with the prior model parametrization. At FI-HYY, the
optimized parameters show higher values of $f_{CA}$ and of $\alpha$ for PFT 15, and of both production parameters ($\alpha$ and
$\beta$) for the dominant PFT (PFT 7). This increase in both soil COS uptake and production after optimization could
correspond to an attempt to better simulate the larger range of variation found in the observations compared to the
modelled fluxes.
Finally, the optimization also affects hydrology-related parameters for both models. However, while it improves
the simulated water content compared to the observations for the mechanistic model at the two sites, it leads to a
degradation at FI-HYY for the empirical model (not shown). Since the empirical model is quite a simplistic model
with few parameters, it relies on parameters from different processes to help better fit the observations – sometimes
degrading the fit to the other processes. The mechanistic model is able to both improve the fit to the COS
observations and soil moisture values implying its parameterization is more consistent.
This optimization experiment has been promising, highlighting how observations can be used to improve the
models. However, since we only optimized over two sites due to the scarcity of soil COS flux observations, for
the global scale simulations in the rest of this study, we will rely on the default parameter values of each
parameterization.
### 3.2   Global scale COS fluxes
#### 3.2.1   Soil COS fluxes
The spatial distribution of oxic soil COS fluxes shows a net soil COS uptake everywhere except in India, in the
Sahel region and some areas in the tropical zone, where net soil COS production is simulated (Figure 7a). The
strongest uptake rates are found in Western North and South America, and in China, with a mean maximum uptake
of -4.4 pmol COS m$^{-2}$ s$^{-1}$ over 2010-2019. The difference in magnitude between the maximum uptake value and
the maximum of production can be noticed, with a net production reaching 67.2 pmol COS m$^{-2}$ s$^{-1}$ in the Sahel
region. India and the Sahel region, where oxic soil COS production is concentrated, are represented in ORCHIDEE
by a high fraction of $C_3$ and $C_4$ crops (Figure S3). In the mechanistic model, crops are associated with the lowest
$f_{CA}$ value due to overall lower fungal diversity and abundance in agricultural fields (Meredith et al., 2019), and the
strongest response of oxic soil COS production to temperature as observed by Whelan et al. (2016). Thus, these
PFT-specific parameters combined with high temperature in the tropical region can explain the net oxic soil COS
production found in these regions. $C_3$ crops are also dominant in China near the Yellow Sea (Figure S3). However,
the mean soil temperature in this region is about 15°C lower than the mean soil temperature in India, leading to a





lower enhancement of soil COS production. The highest atmospheric COS concentration is also found in this
region with about 800 ppt (Figure S2). Indeed, recent inventories have shown that China was related to strong
anthropogenic COS emissions due to the industry, biomass burning, coal combustion, agriculture, or vehicle
exhaust (Yan et al., 2019; Zumkehr et al., 2018). High atmospheric COS concentrations increase soil COS
diffusion and uptake that can compensate part of soil COS production. The highest values of soil COS fluxes for
anoxic soils are located in northern India, with a mean maximum value reaching 36.8 pmol COS $m^{-2}$ $s^{-1}$ (Figure
7b). This region is characterized by rice paddies, which were also associated with strong COS production in
previous studies (Zhang et al., 2004).
The total soil COS fluxes (oxic and anoxic) computed with the mechanistic model (Figure 7c) show a very different
spatial distribution than the one obtained with the empirical model (Figure 7d). Soil COS fluxes from the empirical
model are on the same order of magnitude for net COS uptake than the mechanistic model, with a mean maximum
uptake of -6.41 pmol COS $m^{-2}$ $s^{-1}$. However, most soil COS uptakes simulated by the empirical model is located
in the tropical region, where soil respiration is strong due to high temperature.
The difference of soil COS fluxes between the mechanistic model and the empirical model ranges from -4.1 pmol
COS $m^{-2}$ $s^{-1}$ to +68.0 pmol COS $m^{-2}$ $s^{-1}$ (Appendix D, Figure D1). Over western North and South America, northern
and southern Africa, western Asia, and eastern, northern and Central Asia, the net COS uptake from the
mechanistic model exceeds the uptake from the empirical model. On the contrary, soil COS uptake from the
empirical approach is higher than the net COS uptake simulated with the mechanistic model over Eastern North
and South America, Western, Central and Eastern Africa, and Indonesia. The absence of soil COS production
representation in the empirical approach leads to the strongest differences in India and in the Sahel region, reaching
+68.0 pmol COS $m^{-2}$ $s^{-1}$.
**3.2.2     Temporal evolution of the soil COS budget**
We computed the mean annual soil COS budget over the period 2010-2019 using the monthly variable atmospheric
COS concentration and we compared its evolution to the variations of the mean annual atmospheric COS
concentration.

The evolution of the mean annual soil COS budget (Figure 8) shows small variations in the budget for oxic soils
computed with the mechanistic model between 2010 and 2015, with a net sink ranging from -133 GgS $y^{-1}$ to -124
GgS $y^{-1}$. Then, from 2016 we see a sharp decrease in this budget, which reaches -98 GgS $y^{-1}$ in 2019. This decrease
also corresponds to the decrease in atmospheric COS concentration observed between 2016 and 2019 with a loss
of 25 ppt in 3 years. It is worth noting that other monitoring stations recorded a drop in atmospheric COS
concentration over Europe, as for the GIF station with -42 ppt between 2015 and 2021 (updated after Belviso et
al., 2020). On the contrary, the soil COS net uptake computed with the empirical model slightly increases
from -212 GgS $y^{-1}$ in 2010 to -219 GgS $y^{-1}$ in 2019. As the empirical model defines soils COS flux as proportional
to the total soil respiration independently of atmospheric COS concentration, the budget obtained with this model
is not impacted by the variations observed in atmospheric COS concentration. The anoxic soil COS budget follows
soil temperature variations (not shown), with an increasing trend of about 0.17 GgS $yr^{-1}$ over the studied period.



### 3.3 Transport and site-scale concentrations

**Interhemispheric gradient**

We transported total COS fluxes for the different configurations (i.e. including the soil fluxes but also other components of the COS atmospheric budget, listed in Table 2) with the LMDZ6 atmospheric transport model as described in Sect. 2.1.3. We analyzed COS concentrations derived from simulated COS fluxes obtained with the mechanistic and two empirical approaches with regards to the COS concentrations observed at 14 NOAA sites depicted in Appendix B, Fig. B1. Note that atmospheric mixing ratios of COS result from the transport of all COS sources and sinks and that, due to other sources of errors (transport and errors in the other COS fluxes), the comparison presented in the following should be taken as a sensitivity study of COS seasonal cycle and inter-hemispheric gradient to the soil exchange fluxes rather than a complete validation of one approach or the other. Figure 9: shows the COS atmospheric concentrations at NOAA sites as a function of latitude for each simulated soil flux and for the observations. Here as we want to focus on the latitudinal variations of atmospheric COS mixing ratios, the atmospheric COS concentrations have been vertically shifted to have the same mean as the observations. This means that the concentrations values cannot be compared at each site, we can only compare the interhemispheric gradients of simulated and observed concentrations. The RMSD for the mechanistic model with oxic soils only, the mechanistic model with oxic and anoxic soils, the empirical Berry model (with oxic soils only), and the empirical Launois model (with oxic and anoxic soils) are 36.5, 39.4, 43.0, 51.0 ppt, respectively. While the different approaches show similar gradient patterns in the southern latitudes, they lead to strong differences in the simulated concentrations in the northern hemisphere. Compared to empirical approaches, the mechanistic approach marginally improves the latitudinal distribution of the atmospheric mixing ratios by decreasing the concentrations in the high latitudes. The lower atmospheric mixing ratios above 60 °N reflect the stronger soil absorption in the mechanistic model (see Figure 9), where soil COS uptake is dominant and the compensation by COS production is small (Appendix D, Figure D2). Despite this slight improvement, there are persistent biases as overestimated concentrations at the high latitude sites ALT, BRW, SUM, and underestimated concentrations at most tropical sites: WIS, MLO and SMO. These model-observation mismatches have led top-down studies to identify the missing source as being the tropical oceanic emissions (Berry et al., 2013; Launois et al., 2015; Remaud et al., 2021; Davidson et al., 2021). The present anoxic soil fluxes have little impact on the surface latitudinal distributions and therefore are unlikely to shed new light on the tropical missing source. An explanation for the small impact is that they are located outside areas experiencing deep convection events (e.g. the Indian monsoon domain) and thus the surface concentrations are less sensitive to these fluxes.

**Seasonal cycle at NOAA sites**

Figure 10 shows the detrended temporal evolution of COS concentrations for the mechanistic and empirical approaches at Alert (ALT, Canada) and Harvard Forest (HFM, USA). Because of the mean westerly flow, the HFM site is influenced by continental regions to the west (Sweeney et al., 2015), and is more sensitive to the soil fluxes than other mid-latitude sites located to the west of the ocean (MHD, THD), see Fig. 1 in Remaud et al. (2021). The ALT site samples air masses coming from high-latitude ecosystems (Peylin et al., 1999), but also from regions further south due to atmospheric transport (Parazoo et al., 2011). The reader is referred to Appendix B, Table B2 for the other sites. At both sites, the mechanistic approach tends to weaken the total seasonal amplitude and increase the model-data mismatch. At ALT, the seasonal amplitude is marginally decreased, while at HFM it





is divided by two. At ALT, BRW and SUM, the too high atmospheric concentrations and the too weak seasonal
amplitude given by the mechanistic approach are consistent with an underestimated soil absorption at sites ET-JA
(Estonia) and FI-HYY (Finland) (see Figure 2). As for Harvard Forest, since the mechanistic soil model shows
overall good agreement with the observed soil fluxes (e.g. Figure 2), the model-observation mismatch likely arises
from errors in other components of the COS budget (in particular oceanic and vegetation fluxes). Therefore,
empirical approaches give a more realistic seasonality of atmospheric concentrations for the wrong reasons, which
likely hides an underestimated vegetation uptake. Indeed, as Maignan et al. (2021) showed that the vegetation
uptake magnitude in ORCHIDEE was consistent with measurements, the introduction of variable atmospheric
COS concentrations decreased the vegetation uptake, which as a result, is very likely underestimated now.
Moreover, the comparison between simulated and observed concentrations show a degradation of the simulated
concentrations in this study compared to Maignan et al. (2021). It is to be noted that in addition to using a variable
atmospheric COS concentration in this study, the transported ocean COS fluxes from Masotti et al. (2016) and
Lennartz et al. (2017, 2021) differ from the ones used in Maignan et al. (2021), from Kettle et al. (2002) and
Launois et al. (2015). These results illustrate the necessity of well constraining both the soil and vegetation fluxes
in order to optimize the GPP with the help of atmospheric inverse modelling.
**4    Discussion**
**4.1    Soil budget**
According to the mechanistic approach of this study, the COS budget for oxic soil is a net sink of -126 GgS yr$^{-1}$
over 2009-2016, which is close to the value of -130 GgS yr$^{-1}$ found by Kettle et al. (2002) (Table 5). The
mechanistic model gives the lowest oxic soil COS net uptake compared to all previous studies using empirical
approaches. This budget is also 41% lower than the one found with the Berry empirical approach in this study,
with an uptake of -214 GgS yr$^{-1}$. The anoxic soil COS budget computed with the mechanistic approach is +96 GgS
yr$^{-1}$, which is close to the budget found by Launois et al. (2015) of +101 GgS yr$^{-1}$ based on methane emissions.
However, while COS emissions from anoxic soils were only located in the northern latitudes in Launois et al.
(2015), the COS production in this study is also distributed in the tropical region. Thus, we can expect that despite
similar budget values for anoxic soils, the difference in flux distribution will impact the latitudinal gradient of COS
fluxes. Finally, adding anoxic soil COS budget to oxic soil COS budget results in a total soil COS budget of only
-30 GgS yr$^{-1}$ for the mechanistic approach.
When computing the net total COS budget considering all sources and sinks of COS, the net total from the
empirical approach is closer to zero (-35 GgS yr$^{-1}$) than the net total from the mechanistic model (+149 GgS yr$^{-1}$).
In the empirical approach, neglecting the potential COS production of oxic soils and COS emissions from anoxic
soils leads to a small overestimation of COS sink or underestimation of COS source to close the budget. On the
contrary, the mechanistic approach leads to an overestimation of COS source or an underestimation of COS sink
components. This positive net global budget could be due to an underestimation of vegetation COS uptake in the
northern hemisphere, participating in the underestimation of the COS concentration drawdown (Figure 9), but the
absence of anthropogenic emission seasonality could also play a role. The two net totals obtained in this study are
closer to closing the COS budget than the previous approach from Launois et al. (2015).
Despite a net COS budget closer to zero with the empirical model, it is to be noted that the mechanistic model
better simulates the lack of seasonality at US-HA compared to the empirical model (Figure 2). US-HA is





represented by 80% of PFT6 (temperate broadleaved summergreen forest) and the absence of seasonality by this
PFT has also been reported at a mid-latitude site at Gif-sur-Yvette (Belviso et al., 2020). This PFT is largely found
in the temperate region such as in Europe and in the southern United-States. Moreover, NWR, HFM and LEF
stations are mainly influenced by COS exchanges from the PFT6. Therefore, the use of the mechanistic model
would be recommended to carry out new comparisons at these mid-latitude sites.
**4.2    Variable atmospheric COS concentration versus constant atmospheric COS concentration**
We studied the impacts of using a constant versus a variable atmospheric COS concentration on soil COS fluxes.
At the site-scale we found a distinction between the sites where soil COS production is strong (IT-CRO and ES-
LMA) and the sites mainly showing a net soil COS uptake. The impact of using a constant atmospheric COS
concentration is lower at IT-CRO and ES-LMA because the atmospheric COS concentration does not directly
impact the soil COS production term but participates in the net soil COS flux. Our study shows that at the sites
where a net soil COS uptake is dominant, using a constant atmospheric COS concentration leads to an
underestimation of soil COS flux in winter and an overestimation of soil COS flux from spring to autumn (not
shown). Indeed, during the growing season, plant uptake decreases atmospheric COS concentration (Figure S1),
which reduces COS availability for soil COS diffusion, whereas during winter, a higher atmospheric COS
concentration enhances COS diffusion into the soil.
At the global scale, as the variable atmospheric COS concentration used in this study shows a decrease of about
25 ppt in the recent years (Figure 8), considering a constant atmospheric COS concentration would not enable to
represent the impact of this strong variation on soil COS fluxes. When computing the soil COS budget over 2016
to 2019, we found a net uptake of -126 GgS yr$^{-1}$ with the mechanistic model using a constant atmospheric COS
concentration, against the -110 GgS yr$^{-1}$ computed with a monthly spatially variable concentration. Using a
constant atmospheric COS concentration would then lead to an overestimation of about 13% of the net soil COS
uptake over the past 4 years.
We also studied the impact of considering a constant versus a variable atmospheric COS concentration on the
seasonal variations of mean monthly soil COS fluxes over 2010-2019, simulated with the mechanistic model (not
shown). We found that using a constant atmospheric COS concentration leads to an overestimation of net soil COS
uptake over the whole year in the southern latitudes and from June to February in the northern latitudes (reaching
1.62 pmol m$^{-2}$ s$^{-1}$). This overestimation increases over the regions with the lowest atmospheric COS concentrations,
which limits COS diffusion through the soil matrix. On the contrary when atmospheric COS concentration is high
in the northern latitudes between April and May, considering a constant atmospheric COS concentration leads to
an underestimation of net soil COS uptake. We notice that this underestimation with a constant atmospheric COS
concentration can be found as early as March over Europe, where atmospheric COS concentration is higher in this
region. In eastern Asia, where atmospheric COS concentration is higher than 800 ppt, the underestimation of the
net soil COS uptake can reach -2.34 pmol m$^{-2}$ s$^{-1}$ when considering a constant atmospheric COS concentration.
It is to be noted that the modelled COS concentrations we used have their own uncertainty, which is however
smaller than their difference with the fixed value (Remaud et al., 2021).



### 4.3 Foreseen improvements


The mechanistic representation of soil COS fluxes was found to be in better agreement with the observations at
field sites. However, there can be strong differences between the simulated fluxes and the observations at some
sites, especially at AT-NEU and ES-LMA. In the mechanistic approach, the influence of light on soil COS fluxes
is not considered. Several field studies have reported light-induced emissions in oxic soils (Kitz et al., 2017;
Meredith et al., 2018; Spielmann et al., 2019; Whelan and Rhew, 2015), assumed to be related to the effect of light
on soil organic matter. Spielmann et al. (2019) related strong soil COS emissions during daytime to light at the
sites where direct solar radiations reached the surface, such as ES-LMA and AT-NEU. At these sites, the
mechanistic model was unable to represent the soil COS emission peak during daytime. The optimization we
performed showed that, as expected, adjusting the parameters to site observations improves the fit between the
simulated and observed fluxed. However, it is necessary to represent all important processes in the mechanistic
approach before calibrating the parameters. Thus, a next step in our modelling approach could be to include the
light influence on soil COS fluxes, which can be of major importance for the sites where radiations strongly affect
soil COS fluxes. Mellillo and Steudler (1989) also found that soil COS production could be related to nitrogen
content, which increases with nitrogen fertilizer application. Then crop management practices might also need to
be included when representing the dynamics of soil COS fluxes.
Moreover, one difficulty with the study of soil COS fluxes arises from the scarcity of field measurements that
could be used for data assimilation. Therefore, more field measurements would help to build a larger field
observation database for model validation and calibration. In particular, the characterization of the soil microbial
community should also be addressed to improve the scaling of CA content and activity, represented by the $f_{CA}$
parameter (Meredith et al., 2019).
The mechanistic model from Ogée et al. (2016) has also recently been implemented in the LSM SiB4 (Kooijmans
et al., 2021). In SiB4, the simulated soil COS fluxes with the mechanistic model show a seasonal cycle with a
maximum net soil COS uptake in summer for the sites without crops, while the fluxes computed in ORCHIDEE
show almost no seasonality. The expression of the production term $P$ differs between the two models, which is
based on Meredith et al. (2018) in SiB4 and on Whelan et al. (2016) in ORCHIDEE. The observation sites that are
common to the two studies (FI-HYY, US-HA, AT-NEU and DK-SOR) are also represented by different fractions
of biomes between SiB4 and ORCHIDEE, which changes the parameterization to compute soil COS fluxes.
Finally, the parameter values for the enhancement factor $f_{CA}$ for grass differ as the value for tropical grass is also
assigned to $C_3$ and $C_4$ grass in SiB4. Soil COS flux field data are mainly available in summer, therefore having
field measurements over a whole year could better inform the seasonality of observed soil COS fluxes to compare
to the simulations.
The optimization does not modify the respective seasonality of both soil COS models, with a seasonal cycle that
agrees with the one of soil respiration for the empirical model and a lack of seasonality for the mechanistic model.
The lack of observations in winter does not enable to constrain soil COS fluxes in winter. Therefore, having field
observations over a whole year could help to determine if both models could be calibrated with a constrain over
the whole year instead of only during summer and autumn. Moreover, the optimized set of parameters for the
empirical models leads to a degradation of the simulated soil water content compared to the observations at FI-
HYY, while the optimized parameters of the mechanistic model improve the representation of soil water content





at US-HA and FI-HYY. Thus, the mechanistic approach is to be preferred over the empirical model and should be
selected for future COS studies in ORCHIDEE.
The sensitivity analyses showed the importance of the hydrology-related parameters in the computation of soil
COS fluxes with the mechanistic model. Thus, assuming an accurate representation of soil COS fluxes, soil COS
fluxes could have the potential to add a new constraint on hydrology-related parameters.
In this work, soil COS fluxes are computed in the top 9 cm, which assumes that soil COS uptake and production
depend on the conditions in the first soil layers. Indeed, soil COS uptake depends on diffusive supply of COS from
the atmosphere. However, since soil COS production does not depend on COS supply, deeper soil layers could
also contribute to soil COS production. A study by Yang et al. (2019) presents COS profile measurements in an
orchard, which shows a non-zero COS concentration in deeper soil layers, but no direct evidence for attributing it
to soil COS production. Thus, we could consider deeper soil layers in the future to study the impact on soil COS
fluxes compared to considering only the top soil layers.
The anoxic soil map of regularly flooded wetlands from Tootchi et al. (2019) enables to approximate the spatial
distribution of anoxic soil. However, in our approach, seasonality is only represented through soil temperature
seasonality. Anoxic soil temporal dynamic was initially included in the model described by Ogée et al. (2016) with
the soil redox potential but is not implemented in land surface models such as ORCHIDEE yet. We could also
refine our approach by distinguishing between the different types of wetlands and define a $P_{ref}$ value for each
wetland type instead of a global value of 10 pmol COS m$^{-2}$ s$^{-1}$. Moreover, indirect COS emissions from DMS
oxidation in anoxic soils have been reported (Kettle et al., 2002; Watts, 2000) but are not represented in this study.
Finally, the anoxic map used here represents 9.7% of the global land area, but the distribution of anoxic soils can
greatly vary depending on the study (between 3% and 21%, Tootchi et al., 2019). Therefore, it would also be
interesting to investigate the impact of anoxic soil coverage on soil COS flux uncertainty.
**5  Conclusions and Outlooks**
We have implemented in the ORCHIDEE LSM a mechanistic and an empirical model for simulating soil COS
fluxes. The mechanistic model, that performs a spatialization of the Ogée et al. (2016) model, enables us to
consider that oxic soils can be net COS producers, as illustrated at some of the observation sites. The inter-
hemispheric gradient of COS surface atmospheric mixing ratio is marginally improved when all known COS
sources and sinks are transported with the LMDZ model. This study also highlights the sensitivity of simulated
atmospheric COS concentrations to soil COS flux representation in the northern latitudes. Thus, the uncertainty in
soil COS fluxes could complicate GPP estimation using COS in the northern hemisphere.
The soil COS budget at global scale over the 2009-2016 period is -30 GgS yr$^{-1}$, resulting from the contribution of
oxic soils that represent a net sink of -126 GgS yr$^{-1}$, and of anoxic soils that represent a source of +96 GgS yr$^{-1}$. It
is to be noted that the contribution from anoxic soils, while leading to a similar global budget to Launois et al.
(2015), has a different spatial distribution based on the repartition of regularly flooded wetlands from Tootchi et
al. (2019). This repartition seems more accurate as it also includes anoxic soil COS flux in the tropical region and
considers a larger variety of anoxic soils, such as salt marshes and rice paddies.
During this work, we have also shown the importance of considering spatially and temporally variable atmospheric
COS concentrations on soil COS fluxes, with an especially large impact at global scale. This result evidences the
impact of the recently decreasing atmospheric COS concentrations on the estimated soil COS fluxes.





Regarding the ORCHIDEE model, we performed a sensitivity study highlighting the key parameters to optimize
for the soil models. The impact of soil model parameter optimization was studied at two sites. This study exhibited
strong arguments in favour of the mechanistic model as performing an optimization of the empirical model
parameters can lead to aliasing errors and a degradation of the simulated soil water content. A larger database of
COS flux measurements at the site scale and especially full year time series would greatly help for the next step,
which would be to optimize the parameters of ecosystem COS fluxes.





**Appendix A: Parameters, variables, and constants for soil COS models**

**Table A1: Carbonic anhydrase enhancement factor adapted to ORCHIDEE biomes.**

| ORCHIDEE biomes | **Biomes from** Meredith et al. (2019)Meredith et al. (2019) | $f_{CA}$ **value from** Meredith et al. (2019)Meredith et al. (2019) (unitless) |
|---|---|---|
| 1 - Bare soil | Desert | $13000 \pm 5400$ |
| 2 - Tropical broad-leaved evergreen | Temperate broadleaf forest | $32000 \pm 1800$ |
| 3 - Tropical broad-leaved raingreen | Temperate broadleaf forest | $32000 \pm 1800$ |
| 4 - Temperate needleleaf evergreen | Temperate coniferous forest | $32000 \pm 3100$ |
| 5 - Temperate broad-leaved evergreen | Temperate broadleaf forest | $32000 \pm 1800$ |
| 6 - Temperate broad-leaved summergreen | Temperate broadleaf forest | $32000 \pm 1800$ |
| 7 - Boreal needleleaf evergreen | Temperate coniferous forest | $32000 \pm 3100$ |
| 8 - Boreal broad-leaved summergreen | Temperate broadleaf forest | $32000 \pm 1800$ |
| 9 - Boreal needleleaf summergreen | Temperate coniferous forest | $32000 \pm 3100$ |
| 10 - $C_3$ grass | Mediterranean grassland | $17000 \pm 9000$ |
| 11 - $C_4$ grass | Mediterranean grassland | $17000 \pm 9000$ |
| 12 - $C_3$ agriculture | Agricultural | $6500 \pm 6900$ |
| 13 - $C_4$ agriculture | Agricultural | $6500 \pm 6900$ |
| 14 - Tropical $C_3$ grass | Tropical grassland | $45000$ |
| 15 - Boreal $C_3$ grass | Mediterranean grassland | $17000 \pm 9000$ |







**Table A2: $\alpha$ and $\beta$ parameters for COS production term adapted to ORCHIDEE biomes.**

| ORCHIDEE biomes | Biomes from Whelan et al. (2016) | $\alpha$ parameter from Whelan et al. (2016) (unitless) | $\beta$ parameter from Whelan et al. (2016) (°C⁻¹) |
|---|---|---|---|
| 1 - Bare soil | Desert | N/A | N/A |
| 2 - Tropical broad-leaved evergreen | Rainforest | -8.2 | 0.101 |
| 3 - Tropical broad-leaved raingreen | Rainforest | -8.2 | 0.101 |
| 4 - Temperate needleleaf evergreen | Temperate forest | -7.77 | 0.119 |
| 5 - Temperate broad-leaved evergreen | Temperate forest | -7.77 | 0.119 |
| 6 - Temperate broad-leaved summergreen | Temperate forest | -7.77 | 0.119 |
| 7 - Boreal needleleaf evergreen | Temperate forest | -7.77 | 0.119 |
| 8 - Boreal broad-leaved summergreen | Temperate forest | -7.77 | 0.119 |
| 9 - Boreal needleleaf summergreen | Temperate forest | -7.77 | 0.119 |
| 10 - $C_3$ grass | Savannah | -9.54 | 0.108 |
| 11 - $C_4$ grass | Savannah | -9.54 | 0.108 |
| 12 - $C_3$ agriculture | Soy field | -6.12 | 0.096 |
| 13 - $C_4$ agriculture | Soy field | -6.12 | 0.096 |
| 14 - Tropical $C_3$ grass | Savannah | -9.54 | 0.108 |
| 15 - Boreal $C_3$ grass | Savannah | -9.54 | 0.108 |









**Table A3: Variables for the empirical and mechanistic COS soil models.**

| Variable name | Description | Unit | Reference |
|---|---|---|---|
| Empirical COS soil model | | | |
| $F_{soil,empirical}$ | Empirical model soil COS flux | pmol COS m$^{-2}$ s$^{-1}$ | (Berry et al., 2013) (Yi et al., 2007b) |
| $Resp_{tot}$ | Total (heterotrophic and autotrophic) soil respiration | µmol CO$_2$ m$^{-2}$ s$^{-1}$ | (Yi et al., 2007b) |
| Mechanistic COS soil model | | | |
| $\varepsilon_{tot}$ | Total soil COS porosity | m$^3$ air m$^{-3}$ soil | (Ogée et al., 2016) |
| C | Soil COS concentration | mol m$^{-3}$ | (Ogée et al., 2016) |
| $F_{diff}$ | Soil COS diffusional flux | mol m$^{-2}$ s$^{-1}$ | (Ogée et al., 2016) |
| S | Soil COS consumption rate | mol m$^{-3}$ s$^{-1}$ | (Ogée et al., 2016) |
| P | Soil COS production rate | mol m$^{-3}$ s$^{-1}$ | (Whelan et al., 2016) |
| $F_{soil,mechanistic}$ | Mechanistic model soil COS flux | mol m$^{-2}$ s$^{-1}$ | (Ogée et al., 2016) |
| k | Total COS consumption rate by soil | s$^{-1}$ | (Ogée et al., 2016) |
| B | Solubility of COS in soil water | m$^3$ water m$^{-3}$ air | (Ogée et al., 2016) |
| $\theta$ | Soil volumetric water content | m$^3$ water m$^{-3}$ soil | (Ogée et al., 2016) |
| D | Total effective COS diffusivity in soil | m$^2$ s$^{-1}$ | (Ogée et al., 2016) |
| $z_1$ | Characteristic deep for soil COS flux | m | (Ogée et al., 2016) |
| $k_{uncat}$ | Uncatalysed rate of COS hydrolysis in the soil water | s$^{-1}$ | (Elliott et al., 1989) |
| $k_{cat}$ | Turnover rate of COS enzymatic reaction catalyzed by CA | s$^{-1}$ | (Ogée et al., 2016) |
| $K_m$ | Michaelis-Menten constant of CA catalysis | mol m$^{-3}$ | (Ogée et al., 2016) |
| $x_{CA}$ | Temperature dependence of the ratio $k_{cat}/K_m$ | 1 | (Ogée et al., 2016) |



| k | Soil total COS consumption rate | s$^{-1}$ | (Ogée et al., 2016) |
|---|---|---|---|
| $f_{CA}$ | CA enhancement factor | 1 | (Meredith et al., 2019) |
| D$_{eff,a}$ | Effective diffusivity of gaseous COS in soil | m$^3$ air m$^{-1}$ soil s$^{-1}$ | (Ogée et al., 2016) |
| D$_{eff,l}$ | Effective diffusivity of dissolved COS in soil | m$^3$ water m$^{-1}$ soil s$^{-1}$ | (Ogée et al., 2016) |
| K$_H$ | Henry's law constant | mol m$^{-3}$ Pa$^{-1}$ | (Bird et al., 2002) |
| $D_{0,a}$ | Binary diffusivity of COS in the free air | m$^2$ air s$^{-1}$ | (Bird et al., 2002) |
| $\tau_a$ | Tortuosity factor for gaseous diffusion | 1 | (Ogée et al., 2016) |
| $\tau_{a,r}$ | Tortuosity factor for gaseous diffusion in repacked soils | 1 | (Moldrup et al., 2003) |
| $\tau_{a,u}$ | Tortuosity factor for gaseous diffusion in undisturbed soils | 1 | (Deepagoda et al., 2011) |
| $D_{0,l}$ | Binary diffusivity of COS in the free water | m$^2$ water s$^{-1}$ | (Zeebe, 2011) |
| $\tau_l$ | Tortuosity factor for solute diffusion | 1 | (Millington and Quirk, 1961) |
| $\alpha$ | COS production parameter | 1 | (Whelan et al., 2016) |
| $\beta$ | COS production parameter | 1 | (Whelan et al., 2016) |
| ORCHIDEE LSM | | | |
| p | Pressure | | ORCHIDEE LSM |
| $\varepsilon_a$ | Air-filled porosity | m$^3$ air m$^{-3}$ soil | ORCHIDEE LSM |
| $\varphi$ | Total soil porosity (air-filled and water-filled pores) | m$^3$ m$^{-3}$ | ORCHIDEE LSM |
| T | Mean soil temperature | K | ORCHIDEE LSM |



| t | time | s | ORCHIDEE LSM |
|---|------|---|--------------|
| z | depth | m | ORCHIDEE LSM |






**Table A4: Constants for the empirical and mechanistic COS soil models.**

| Constant name | Description | Value | Unit | Reference |
|---|---|---|---|---|
| Empirical COS soil model | | | | |
| $k_{soil}$ | Constant to converts $CO_2$ production from respiration to a COS uptake | 1.2 | pmol COS/µmol $CO_2$ | (Yi et al., 2007) |
| Mechanistic COS soil model | | | | |
| $C_a$ | Ambient air COS concentration when chosen constant (500 ppt) | $2.0437 \times 10^{-8}$ | mol m$^{-3}$ | |
| $z_{max}$ | Maximum soil depth | 0.09 | m | ORCHIDEE LSM |
| $pK_w$ | Dissociation constant of water | 14 | 1 | |
| $\Delta H_a$ | Thermodynamic parameter | 40 | kJ mol$^{-1}$ | (Ogée et al., 2016) |
| $\Delta H_d$ | Thermodynamic parameter | 200 | kJ mol$^{-1}$ | (Ogée et al., 2016) |
| $\Delta S_d$ | Thermodynamic parameter | 660 | J mol$^{-1}$ K$^{-1}$ | (Ogée et al., 2016) |
| R | Ideal gas constant | 8.314 | J mol$^{-1}$ K$^{-1}$ | |
| $D_{0,a}(25°C, 1\,atm)$ | Binary diffusivity of COS in the free air at 25°C and 1 atm | $1.27 \times 10^{-5}$ | m$^2$ s$^{-1}$ | (Massman, 1998) |
| $D_{0,l}(25°C)$ | Binary diffusivity of COS in the free water at 25°C | $1.94 \times 10^{-9}$ | m$^2$ s$^{-1}$ | (Ulshöfer et al., 1996) |
| $Q_{10}$ | Multiplicative factor of the production rate for a 10 °C temperature rise | 2.7 | 1 | (Meredith et al., 2018) |
| $P_{ref}$ | Reference production term | 10 | pmol m$^2$ s$^{-1}$ | |






**Appendix B: Locations and descriptions of the observation sites**

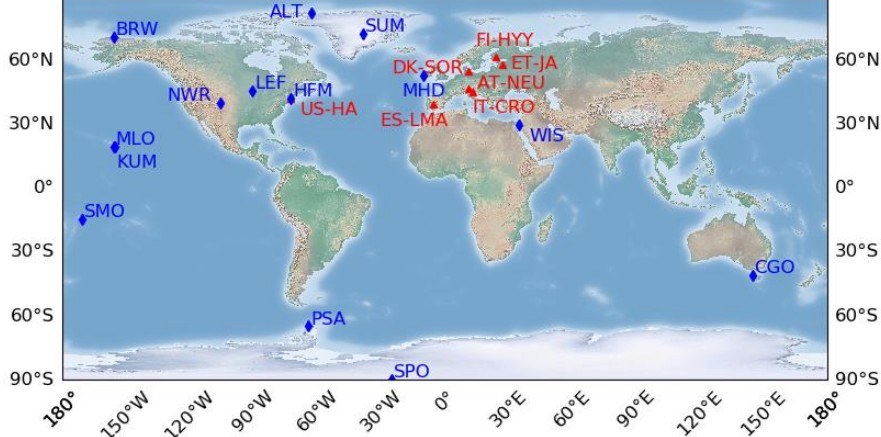

**Figure B1: Locations of the observation sites for soil COS flux measurements (red) and atmospheric concentration**
**measurements (blue).**
**Table B1: List of air sampling sites selected for evaluation of COS concentrations.**

| Site | Short name | Coordinates | Elevation (m above sea level) | Comments |
|---|---|---|---|---|
| South Pole, Antarctica, United States | SPO | 90.0°S, 24.8°E | 2810 | |
| Palmer Station, Antarctica, United States | PSA | 64.77°S, 64.05°W | 10.0 | |
| Cape Grim, Australia | CGO | 40.68°S, 144.69°E | 164 | inlet is 70 m aboveground |
| Tutuila, American Samoa | SMO | 14.25°S, 170.56°W | 77 | |
| Mauna Loa, United States | MLO | 19.54°N, 155.58°W | 3397 | |
| Cape Kumukahi, United States | KUM | 19.74°N, 155.01°W | 3 | |
| Weizmann Institute of Science at the Arava Institute, Ketura, Israel | WIS | 29.96°N, 35.06°E | 151 | |
| Niwot Ridge, United States | NWR | 40.04°N, 105.54°W | 3475 | |
| Harvard Forest, United States | HFM | 42.54°N, 72.17°W | 340 | inlet is 29 m aboveground |
| Wisconsin, United States | LEF | 45.95°N, 90.28°W | 868 | inlet is 396 m aboveground on a tall tower |





| Mace Head, Ireland | MHD | 53.33°N, 9.9°W | 18 | |
| Barrow, United States | BRW | 71.32°N, 155.61°W | 8 | |
| Summit, Greenland | SUM | 72.6°N,38.42°W | 3200 | |
| Alert, Canada | ALT | 82.45°N, 62.51°W | 195 | |



**Table B2: Normalized standard deviations (NSDs) of the simulated concentrations by the observed concentrations.**
**Within brackets are the Pearson correlation coefficients (r) between simulated and observed COS concentrations for**
**the mechanistic and empirical approaches, calculated between 2011 and 2015 at selected NOAA stations. For each**
**station, NSD and r closest to one are in bold and farthest ones are in italic. The time-series have been detrended**
**beforehand and filtered to remove the synoptic variability (see Sect. 2.3.3).**

| | SMO | KUM | MLO | NWR | LEF | HFM | MHD | SUM | BRW | ALT |
|---|---|---|---|---|---|---|---|---|---|---|
| Mechanistic | 1.1 | 0.7 | **0.9** | *0.4* | *0.2* | *0.3* | *1.5* | 0.4 | **1.1** | *0.8* |
| (Oxic) | **(0.8)** | (0.7) | (0.8) | *(0.4)* | (0.7) | (0.8) | (0.2) | (0.2) | *(0.1)* | *(0.1)* |
| Empirical | **1.0** | 0.8 | 1.2 | **0.8** | 0.5 | 0.6 | 1.5 | 0.5 | *1.3* | **0.9** |
| (Oxic) | (0.7) | **(0.9)** | **(0.9)** | (0.4) | **(0.9)** | **(0.9)** | **(0.4)** | (0.6) | (0.3) | **(0.4)** |
| Mechanistic | *1.2* | *0.6* | **0.9** | 0.5 | *0.2* | *0.3* | **1.0** | *0.4* | *1.3* | *0.8* |
| (Oxic+Anoxic) | (0.7) | *(0.6)* | (0.7) | (0.1) | *(0.2)* | (0.5) | *(0.1)* | *(0.0)* | *(0.1)* | *(0.1)* |
| Launois | 1.1 | **1.0** | *1.4* | 1.4 | **0.9** | **0.8** | 1.6 | **0.6** | 1.2 | **0.9** |
| (Oxic+Anoxic) | *(0.6)* | **(0.9)** | **(0.9)** | **(0.7)** | **(0.9)** | **(0.9)** | **(0.4)** | **(0.7)** | **(0.4)** | **(0.4)** |





**Appendix C: Soil COS production term for the mechanistic model**

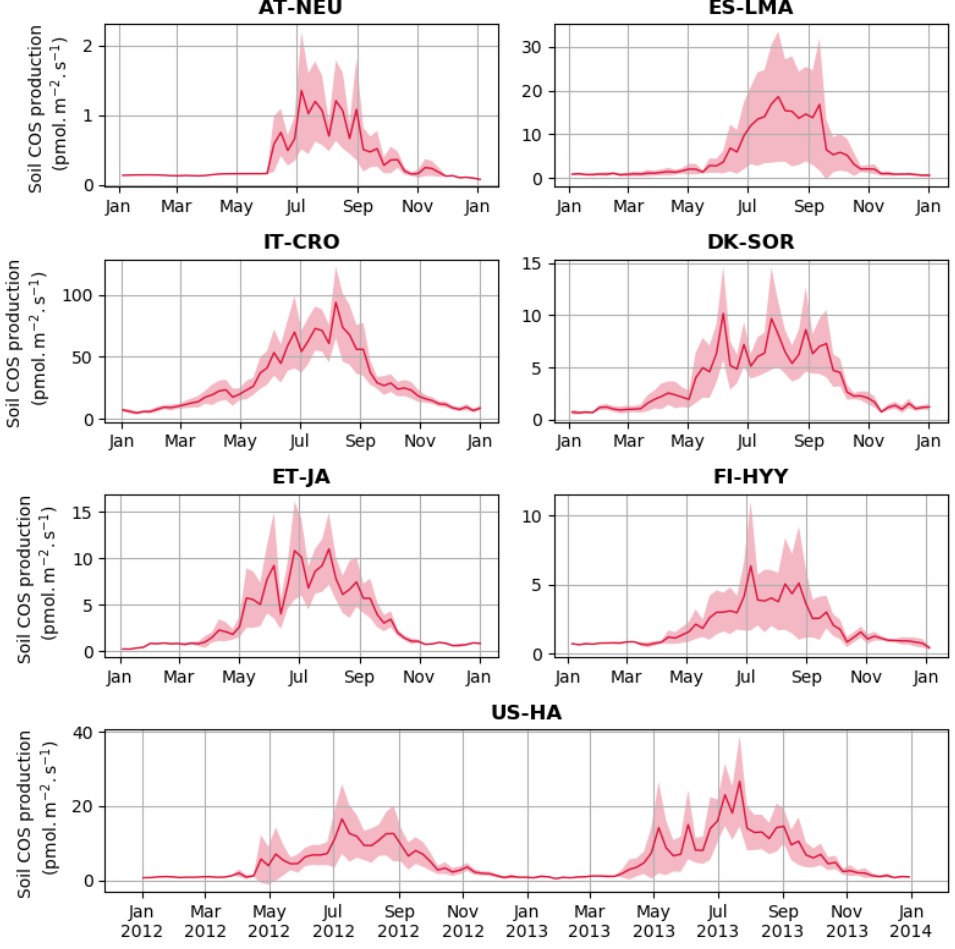


**Figure C1: Seasonal cycles of soil COS production with weekly average production at AT-NEU, ES-LMA, IT-CRO,**
**DK-SOR, ET-JA, FI-HYY, US-HA. The shaded areas above and below the modelled curve represent the standard-**
**deviation over a week. Soil COS production was computed with a variable atmospheric COS concentration.**






**Appendix D: Global scale soil COS fluxes**

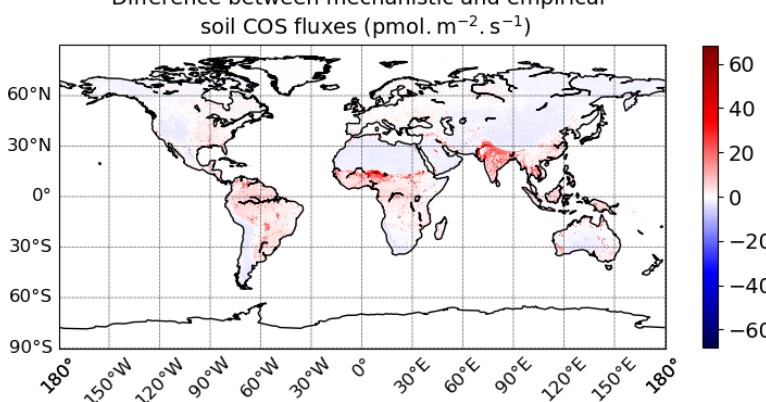

**Figure D1: Mean difference between soil COS fluxes computed with the mechanistic and the empirical model over 2010-**
**2019. The map resolution is 0.5°x0.5°.**

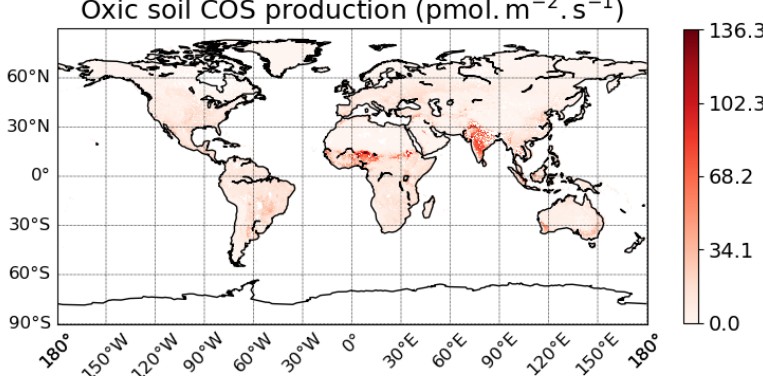

**Figure D2: Mean spatial distribution of oxic soil COS production term over 2010-2019. The map resolution is 0.5°x0.5°.**











**Appendix E: Prior versus post optimization parameter values**

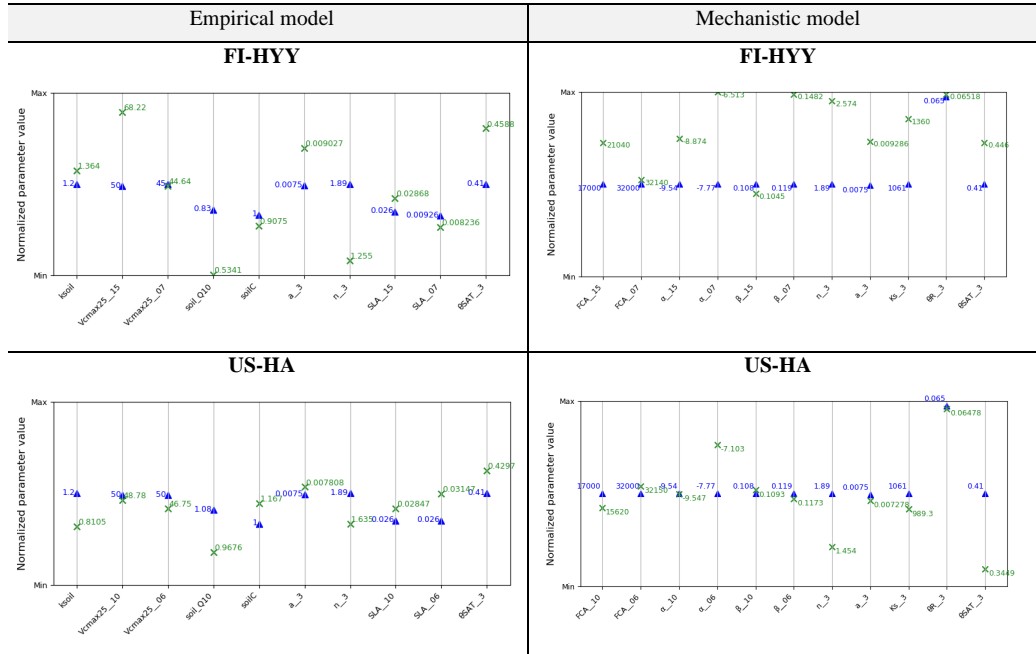

**Figure E1: Comparison between prior and posterior optimization parameter values at FI-HYY and US-HA. The y-axis represents the normalization between the edges of the range of variation for each parameter.**



*Code availability.* The CMIP6 version of the ORCHIDEE model including the soil COS sub-models is available
on request to the authors. The LMDZ model is available from http://web.lmd.jussieu.fr/LMDZ/LMDZ6/ (last
access: 21 October 2021) under the CeCILL v2 Free Software License.

Data availability. For FI-HYY, we used the 2015 soil chamber COS measurements published in Sun et al. (2018).
For US-HA, we used the soil COS flux data derived from eddy covariance COS and $CO_2$ measurements and soil
chamber $CO_2$ measurements conducted in 2012 and 2013, published in Wehr et al. (2017). We used the COS flux
data published in Kitz et al. (2020) and Spielmann et al. (2019) for AT-NEU in 2015, DK-SOR and ES-LMA in
2016 and IT-CRO in 2017.

*Author contributions.* CA, FM, MR, and PP conceived the research. JO advised regarding the spatialization of his
mechanistic model. CA and FM coded the ORCHIDEE developments and made the simulations. MR transported
all COS sinks and sources with the LMDZ model. FK, FMS, and GW provided the data for AT-NEU, ES-LMA,
DK-SOR, IT-CRO and ET-JA. WS provided the data for FI-HYY site and RW for the US-HA site. NR provided
code and guidance for the sensitivity analysis and data assimilation experiments. SB, JEC, MEW, DH, STL, US
and DM were consulted on their respective expertise.

*Competing interests.* The authors declare that they have no conflict of interest.

*Acknowledgments.*
The authors are very grateful to everyone who participated in field data collection used in this study. We thank
Vladislav Bastrikov for providing the ORCHIDAS code. We also acknowledge Nicolas Vuichard for providing
the soil bulk density map used in ORCHIDEE simulations. Operation of the US-HA site is supported by the
AmeriFlux Management Project with funding by the U.S. Department of Energy's Office of Science under
Contract No. DE-AC02-05CH11231 and additionally is a part of the Harvard Forest LTER site supported by the
National Science Foundation (DEB-1832210). The field campaign at DK-SOR was supported by the Danish ICOS
contribution (ICOS/DK) and by the Danish Council for Independent Research grant DFF-1323-00182.

*Financial support.*
This research has been mainly supported by the European Commission, Horizon 2020 Framework Programme,
4C (grant no. 821003) and to a small extend VERIFY (grant no. 776810)
FK, FMS and GW acknowledge funding by the Austrian National Science Fund (FWF) through contracts P26931,
P27176, P31669 and I03859 and the University of Innsbruck.



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




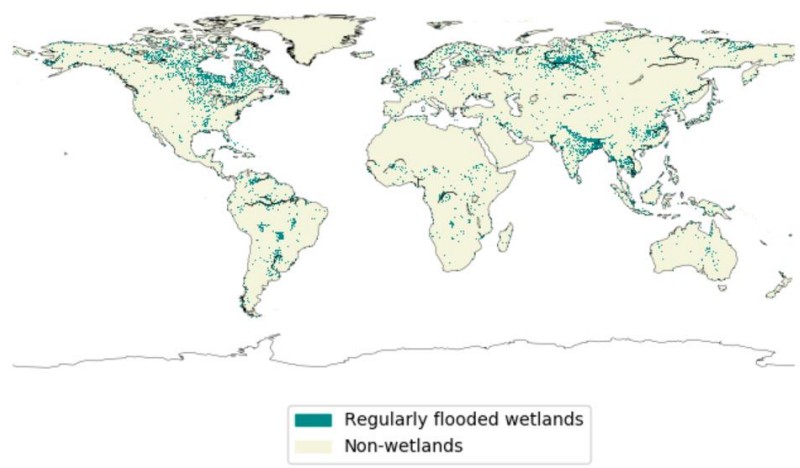


**Figure 1: Map of wetlands distribution used to represent anoxic soils in ORCHIDEE. The map resolution is 0.5°x0.5° (adapted from Tootchi et al., 2019).**






**Table 1: lists the sites' characteristics including their identification name, location, climate, soil type, dominant**
**vegetation and species, corresponding PFT fractions we used for the ORCHIDEE simulations, and reference studies for**
**more details. The spatial distribution of the sites is represented in Appendix B, Figure B1.**

| | Grassland | Savannah-like grassland | Deciduous broadleaf forest | Agricultural soybean field | Evergreen needleleaf forest | Boreal evergreen needleleaf forest | Temperate deciduous broadleaf forest |
|---|---|---|---|---|---|---|---|
| Country | Austria | Spain | Denmark | Italy | Estonia | Finland | United-States |
| Sampling site | Neustift | Las Majadas del Tietar | Sorø | Rivignano | Järvselja | Hyytiälä | Harvard |
| ID | AT-NEU | ES-LMA | DK-SOR | IT-CRO | ET-JA | FI-HYY | US-HA |
| Coordinates | 47°07′N, 11°19′E | 39°56′N, 5°46′W | 55°29′N, 11°38′E | 45°52′N, 13°05′E | 58°16′N, 27°18′E | 61.85°N, 24.29°E | 42.54°N, 72.17°W |
| Climate | Humid continental | Mediterranean | Temperate maritime | Humid subtropical | Temperate | Boreal | Cool, moist temperate |
| Soil type | Fluvisol | Abruptic Luvisol | Alfisols or Mollisols | Silt loam | Haplic Gleysol | Haplic podzol | Sandy loam glacial till |
| Dominant vegetation | Graminoids: *Dactylis glomerata, Festuca pratensis* Forbs: *Ranunculus acris, Taraxacum officinale* | Tree: *Quercus ilex* Grass: *Vulpia bromoides* | European beech (*Fagus sylvatica*) | Soybean | Norway spruce (*Picea abies*) | Scots pine (*Pinus sylvestris*) | Red oak (*Quercus rubra*), Red maple (*Acer rubrum*), Hemlock (*Tsuga canadensis*). |
| ORCHIDEE PFT representation | 100% temperate natural grassland ($C_3$) (PFT 10) | 20% temperate broadleaf evergreen (PFT 5) 80% temperate natural grassland ($C_3$) (PFT 10) | 80% boreal broadleaf summergreen (PFT 8) 20% boreal natural grassland ($C_3$) (PFT 15) | 100% C3 crops (PFT 12) | 50% boreal needleleaf evergreen (PFT 7) 40% boreal broadleaf summergreen (PFT 8) 10% boreal natural grassland ($C_3$) (PFT 15) | 80% boreal needleleaf evergreen (PFT 7) 20% boreal natural grassland ($C_3$) (PFT 15) | 80% temperate broadleaf summergreen (PFT 6) 20% of temperate natural grassland ($C_3$) (PFT 10) |
| References | Hörtnagl et al. (2011) Hörtnagl and Wohlfahrt (2014) Spielmann et al. (2019) Kitz et al. (2020) | Lopez-Sangil et al. (2011) El-Madany et al. (2018) Weiner et al. (2018) Spielmann et al. (2019) Kitz et al. (2020) | Pilegaard et al. (2011) Wu et al. (2013) Brændholt et al. (2018) Spielmann et al. (2019) Kitz et al. (2020) | Spielmann et al. (2019) | Noe et al. (2011, 2015) Kitz et al. (2020) | Kolari et al. (2009) Sun et al. (2018) | Urbanski et al. (2007) Wehr et al. (2017) |






**Table 2: Prescribed COS surface fluxes used as model input. Mean magnitudes and standard deviations of different**
**types of fluxes are given for the period 2009-2016.**

| Type of COS flux | Temporal resolution | Total ($Gg\ S\ yr^{-1}$) | Standard deviation ($Gg\ S\ yr^{-1}$) | Data Source |
|---|---|---|---|---|
| Anthropogenic | Monthly, interannual | 394 | 21 | Zumkehr et al. (2018). The fluxes for the year 2012 were repeated after 2012. |
| Biomass burning | Monthly, interannual | 48 | 9 | Stinecipher et al. (2019) |
| Soil | Monthly, interannual | See Table 5. | 5 (oxic) 2 (anoxic) | This work, including mechanistic and empirical approaches (Berry et al., 2013; Launois et al., 2015) |
| Ocean | Monthly, interannual | 313 | 14 | Lennartz et al. (2021) and Masotti et al. (2015) for indirect oceanic emissions (via CS2 and DMS respectively), and Lennartz et al. (2017) for direct oceanic emissions |
| Vegetation uptake | Monthly, interannual | -576 | 7 | Maignan et al. (2021) |



**Table 3: Comparison between simulated and measured weekly soil COS fluxes (RMSD) at ES-LMA, DK-SOR, IT-**
**CRO, AT-NEU, ET-JA, FI-HYY and US-HA. Values in bold show the highest accuracy between modelled and**
**measured soil COS fluxes for each site (smallest RMSD values). Soil COS fluxes are computed with a variable**
**atmospheric COS concentration.**

|  | Empirical model | Mechanistic model |
|---|---|---|
| ES-LMA | **3.20** | 4.39 |
| DK-SOR | **0.45** | 0.75 |
| IT-CRO | 4.45 | **2.58** |
| AT-NEU | **1.76** | 1.82 |
| ET-JA | 0.84 | **0.46** |
| FI-HYY chamber 1 | 1.21 | **0.89** |
| FI-HYY chamber 2 | 0.99 | **0.58** |
| US-HA | 3.34 | **1.19** |
| Mean all sites | 2.03 | **1.58** |


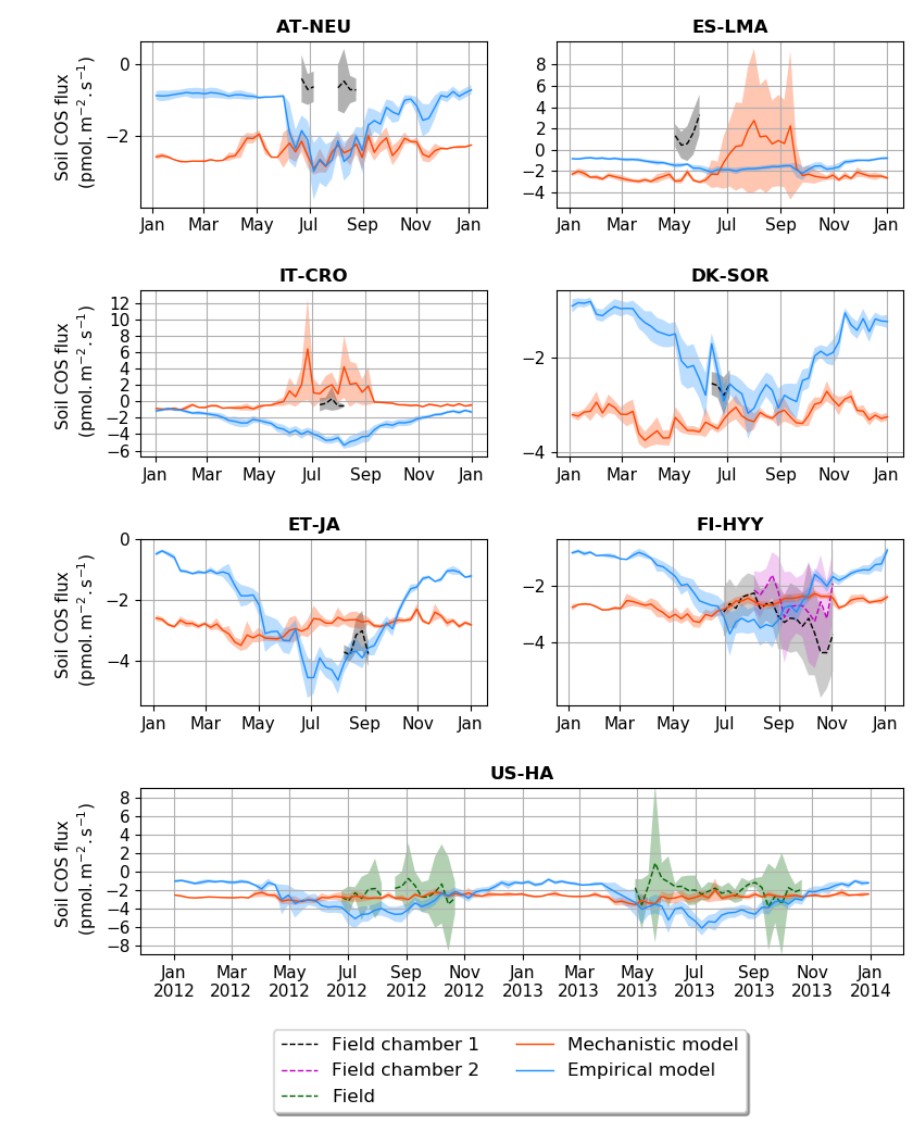


**Figure 2: Seasonal cycle of weekly average net soil COS fluxes (pmol m⁻² s⁻¹) at: AT-NEU, ES-LMA, IT-CRO, DK-SOR, ET-JA, FI-HYY and US-HA. The shaded areas around the observation and simulation curves represent the standard-deviation over a week for each site. Soil COS fluxes are computed with a variable atmospheric COS concentration.**










**Table 4: Comparison between simulated and measured half-hourly soil COS fluxes (RMSD) at ES-LMA, DK-SOR, IT-**
**CRO, AT-NEU, ET-JA, FI-HYY and US-HA. Values in bold show the highest accuracy between modelled and**
**measured soil COS fluxes for each site (smallest RMSD values). Soil COS fluxes are computed with a variable**
**atmospheric COS concentration.**

|  | Empirical model | Mechanistic model |
|---|---|---|
| ES-LMA | **2.71** | 3.90 |
| DK-SOR | **0.39** | 0.91 |
| IT-CRO | 3.82 | **1.33** |
| AT-NEU | **2.22** | 2.24 |
| ET-JA | **0.22** | 1.21 |
| FI-HYY chamber 1 | 0.97 | **0.18** |
| FI-HYY chamber 2 | 1.39 | **0.73** |
| US-HA | 3.21 | **0.54** |
| Mean all sites | 1.87 | **1.38** |


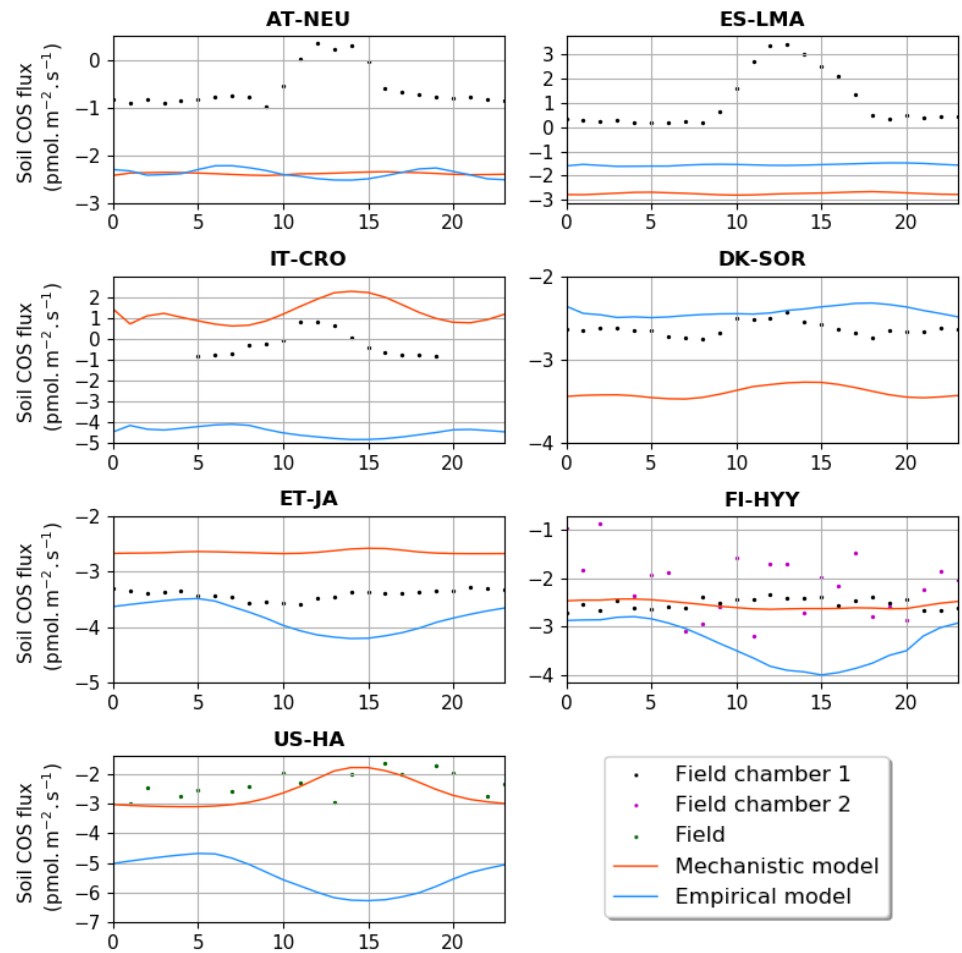

**Figure 3: Mean diel cycle of net soil COS fluxes (pmol m$^{-2}$ s$^{-1}$) over a month at: AT-NEU (08/2015), ES-LMA (05/2016), IT-CRO (07/2017), DK-SOR (06/2016), ET-JA (08/2016), FI-HYY (08/2015) and US-HA (07/2012). Soil COS fluxes are computed with a variable atmospheric COS concentration. The observation-based diel cycles (dots) are computed using Random Forest models at At-NEU, ES-LMA, IT-CRO, DK-SOR and ET-JA. At AT-NEU and ES-LMA.**

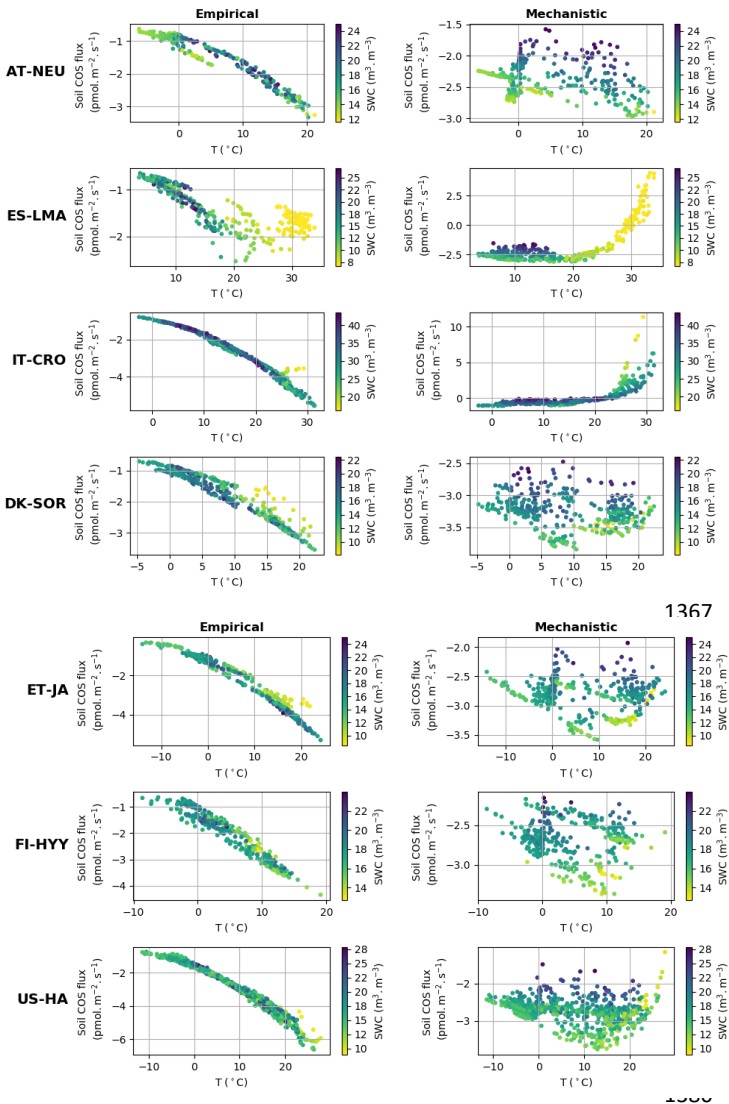

1381

**Figure 4: Simulated daily average net soil COS flux (pmol m² s⁻¹) versus soil temperature (°C) and soil water content (SWC) (m³.m⁻³) at AT-NEU, ES-LMA, IT-CRO, DK-SOR, ET-JA, US-HA and FI-HYY, for the empirical and the mechanistic model.**

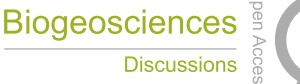



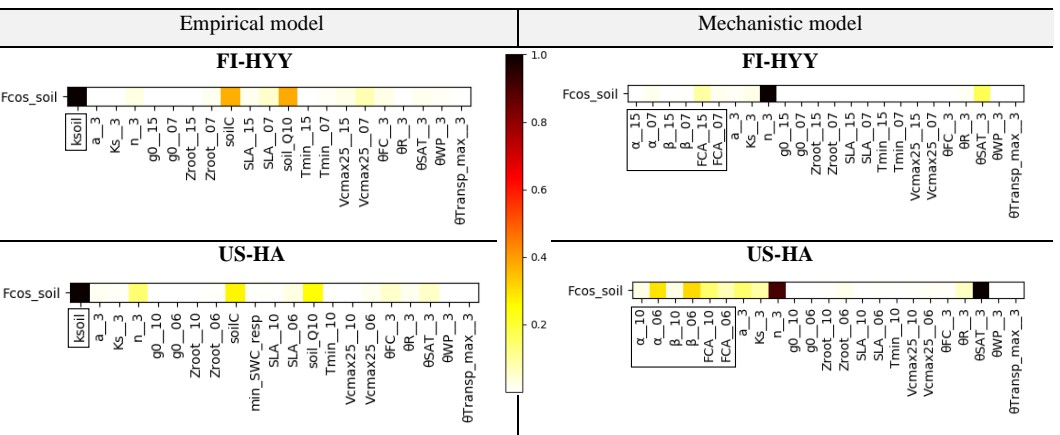

1385

**Figure 5: Morris sensitivity scores of the key parameters to which soil COS fluxes are sensitive, for the empirical (left) and the mechanistic (right) models. The two studied sites are FI-HYY (top) and US-HA (bottom). Full descriptions of each tested parameter can be found in Tables S3 and S4 in the supporting information. The numbers at the end of the parameter names correspond to the PFTs at each site for the PFT-dependent parameters, and to the dominant soil texture for soil texture-dependent parameters (soil texture number 3, i.e. sandy loam, at FI-HYY and US-HA). The first-order parameters are shown in the frames.**



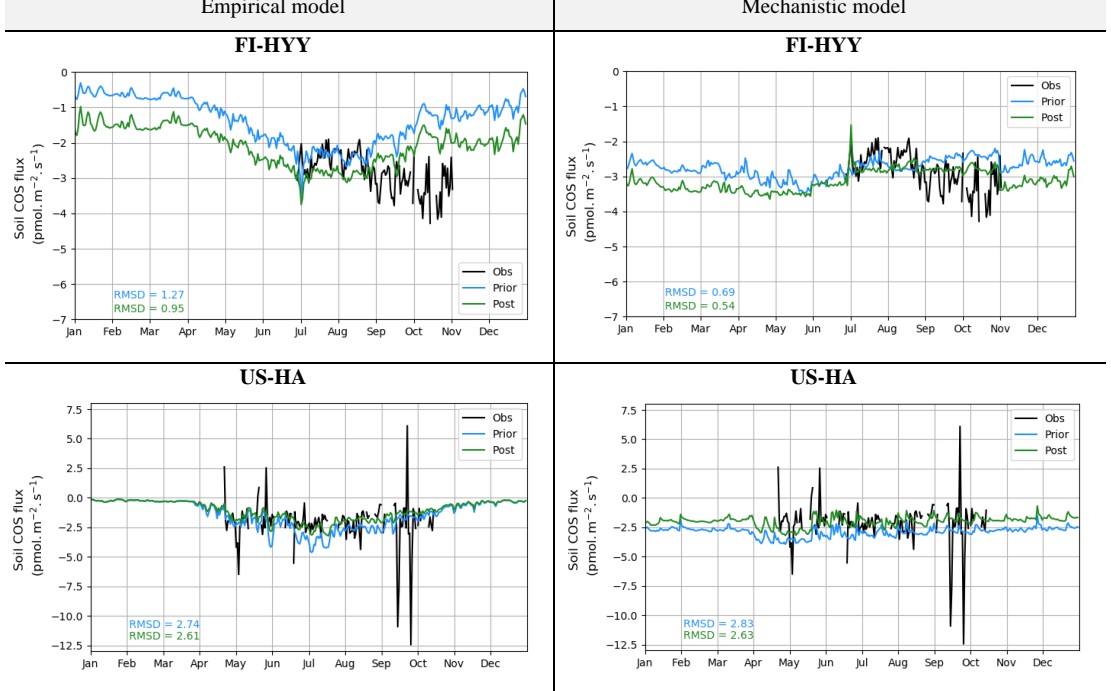


**Figure 6: Prior and post optimization net soil COS fluxes (pmol m$^{-2}$ s$^{-1}$) for the empirical (left) and the mechanistic (right) models. The two studied sites are FI-HYY (top) in 2015 and US-HA (bottom) in 2013.**




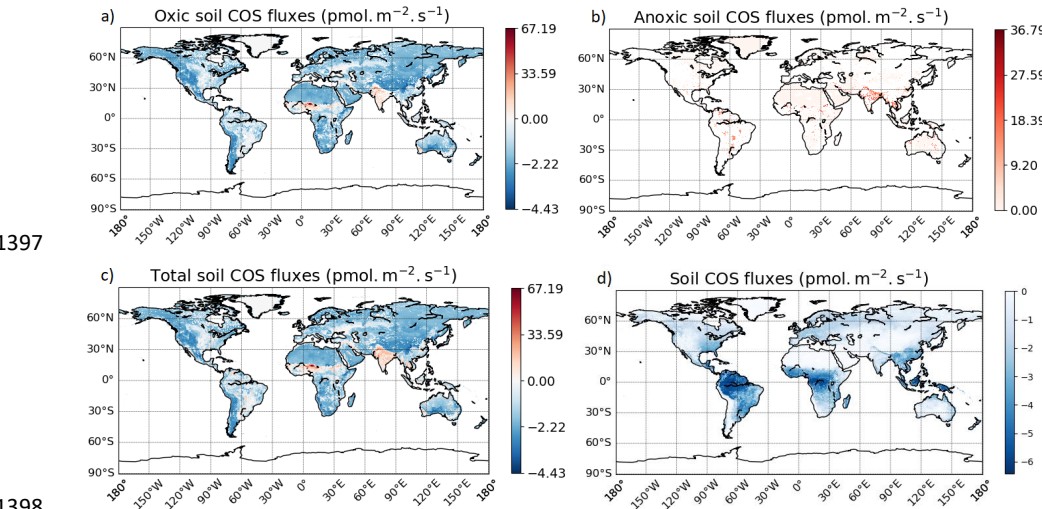



**Figure 7: Maps of mean soil COS fluxes for the mechanistic (a, b, c) and the empirical model (d), computed over 2010-**
**2019 with a variable atmospheric COS concentration. Color scales were normalized between the minimum and**
**maximum soil COS flux values and centered on zero for oxic and total soil COS fluxes computed with the mechanistic**
**model. The map resolution is 0.5°x0.5°.**

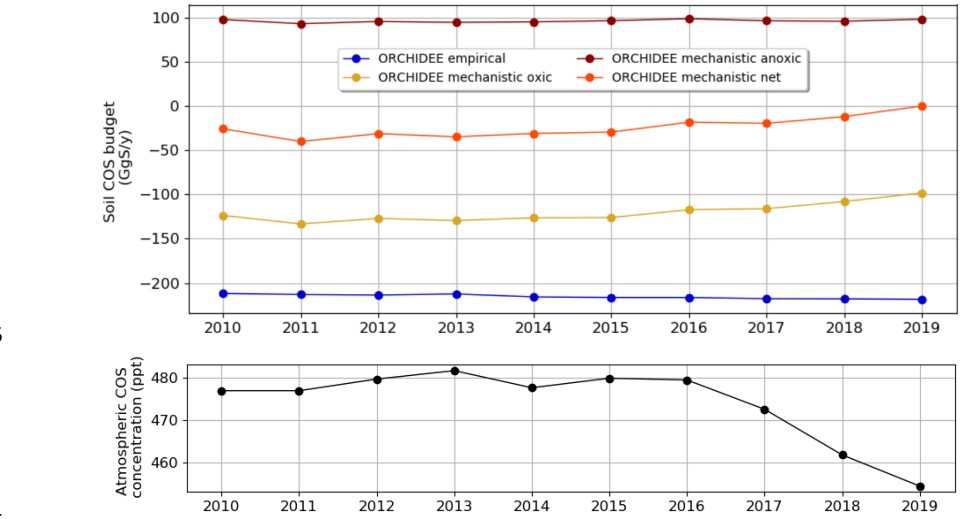


**Figure 8: Evolution of mean annual soil COS budget and mean annual atmospheric COS concentration between 2010**
**and 2019, computed with a variable atmospheric COS concentration.**


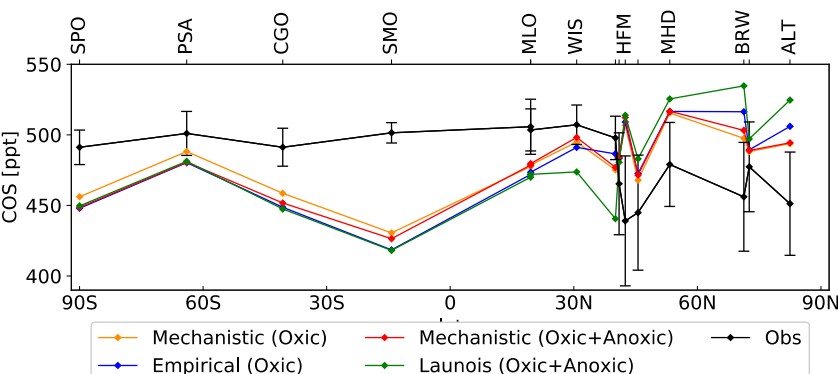


**Figure 9: Comparison of the latitudinal variations of the COS abundances simulated by LMDZ at NOAA sites with the observations (black). The LMDZ COS abundances have been vertically shifted such that the means of the simulated concentrations are the same as the mean of the observations. The error bars around the black curve represent the standard deviation over the whole studied period at each NOAA site. The orange curve is obtained using the oxic soil fluxes of the mechanistic model. The red curve is obtained using the oxic and anoxic soil fluxes of the mechanistic model. The blue curve is given by LMDZ using the oxic soil fluxes from the Berry empirical model. The green curve is obtained using the soil fluxes from the empirical approach of Launois et al. (2015). For more clarity, the names of the stations KUM (19.74°N, 155.01°W), NWR (40.04°N, 105.54°W), LEF (45.95°N, 90.28°W) and SUM (72.6°N,38.42°W) are not shown on this figure due to their proximity to other stations (Appendix B, Figure B1 and Table B1).**

1420

1421

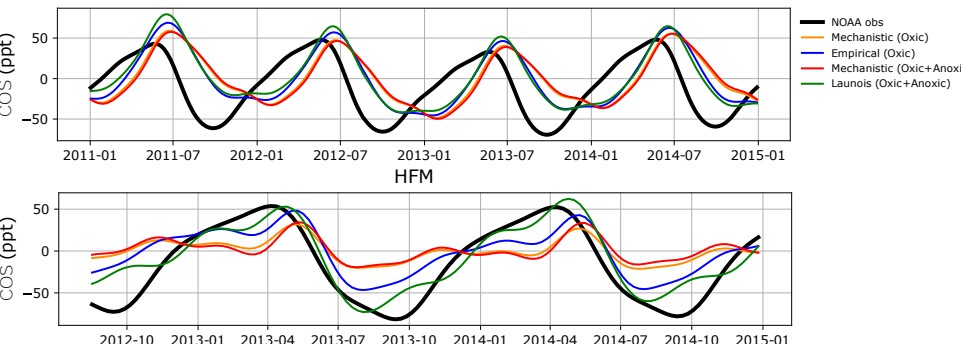

1422

**Figure 10. Detrended temporal evolution of simulated and observed COS concentrations at two selected sites, for the mechanistic (Oxic soils alone, and Oxic + Anoxic soils) and empirical approaches (Berry et al., 2013; Launois et al., 2015) simulated with LMDZ6 transport between 2011 and 2015. Top: Alert station (ALT, Canada), bottom: Harvard Forest station (HFM, USA). The curves have been detrended beforehand and filtered to remove the synoptic variability (see Sect. 2.3.3).**






**Table 5: Comparison of soil COS budget per year (GgS yr$^{-1}$). The net total COS budget is computed by adding all**
**sources and sinks of COS used to transport COS fluxes (Table 2).**

| | Kettle et al. (2002) | Berry et al. (2013) | Launois et al. (2015) | | | This study | |
|---|---|---|---|---|---|---|---|
| | | | ORCHIDEE | LPJ | CLM4 | Empirical soil model | Mechanistic soil model |
| Period | 2002 | 2002–2005 | 2006-2009 | | | 2009-2016 | |
| Plants | -238 | -738 | -1335 | -1069 | -930 | -576 | |
| Soil oxic | -130 | -355 | -510 | | | -214 | -126 |
| Soil anoxic | +26 | Neglected | +101 | | | Neglected | +96 |
| Soil total | -104 | -355 | -409 | | | -214 | -30 |
| Net total | +64 | +1 | -566 | -300 | -161 | -35 | +149 |

