# Peer review of "Global modelling of soil carbonyl sulfide exchanges 1"

_Biogeosciences, 2021_

## Author Comment (AC1)

Biogeosciences Discuss., referee comment RC1 https://doi.org/10.5194/bg-2021-281-RC1, 2021 © Author(s) 2021. This work is distributed under the Creative Commons Attribution 4.0 License.

**Comment on bg-2021-281**

Anonymous Referee #1

Referee comment on "Global modelling of soil carbonyl sulfide exchange" by Camille Abadie et al., Biogeosciences Discuss., https://doi.org/10.5194/bg-2021-281-RC1, 2021

Abadie et al. implemented a mechanistic and empirical soil model of COS exchange into the ORCHIDEE land surface model and compare those with observations of soil COS fluxes at several sites, representing different soil types. Through a sensitivity study they find the most important parameters for the soil COS flux and optimize those parameters with observations at two sites to improve the COS soil flux simulations. Finally, the authors provide an updated global soil COS budget, including both oxic and anoxic (wetland) soils. This is a very complete and thorough study that I find very well readable. My comments are hence minor.

Answer: The authors thank the Referee for the overall positive answer to this study and for the very helpful comments to improve the manuscript.

Abstract:

P1, L44: -576 Gg S yr-1 for vegetation+soil, or only the vegetation?

Answer: -576 GgS yr-1 corresponds to the revised budget for vegetation only. We replaced "which helped reduce the imbalance of the atmospheric COS budget by lowering COS uptake by soils and vegetation globally (-10% for soil, and -8% for vegetation with a revised mean estimate of -576 GgS y-r1 over 2009-2016)" by "which helped reduce the imbalance of the atmospheric COS budget by lowering **soil COS uptake by 10% and plant COS uptake by 8% globally (with a revised mean vegetation budget of -576 GgS yr-1 over 2009-2016)".**

Introduction:

P2, L62-63: This sentence reads weird.

Answer: We rephrased this sentence as shown in the answer of the next comment.

*P2*, L63-64: The numbers 700-1100 GgS yr-1 sound like a very large gap. If I'm correct, Berry et al. (2013) added an additional ocean flux of 600 to close the gap, and Kuai et al. (2015) added 559 Gg S yr-1. Do the values 700-1100 GgS yr-1 represent the total ocean emissions? So not only the emission gap? A reference to a more recent inversion could be added (Ma et al. (2021) with a total gap of 432Gg S yr-1).

Answer: Indeed, the values 700-1100 GgS yr-1 represent the total COS oceanic emission

estimates (Berry et al., 2013; Kuai et al., 2015; Launois et al, 2015). We modified the sentence as follows "Several atmospheric transport inversion studies have suggested that an unidentified COS source located over the tropics, of the order of 400-600 GgS yr-1, was needed to close the contemporary COS budget (Berry et al., 2013; Glatthor et al., 2015; Kuai et al., 2015; Ma et al., 2021; Remaud et al., 2022)."

*P2, L78: better say something like: "they have usually not been considered in atmospheric COS budgets".*

Answer: We replaced the sentence as suggested by the Referee "Although such COS emissions can be large in some conditions, **they have usually not been considered in atmospheric COS budgets**."

*P3, L87: form = from*

Answer: Thank you, this error was corrected.

*P3, L113: For clarification, consider adding something like:* "at the different sites **that will be used for evaluation in this study**".

Answer: We completed the sentence as suggested "To better represent the observed soil conditions at the different sites **that will be used for evaluation in this study**, we substituted the soil textures initially assigned in ORCHIDEE from the USDA texture global map with the observed soil textures corresponding to the USDA texture classes (Table S2)."

**P3, L119: More important to the mechanistic model than to the empirical model?**

Answer: Yes, this precision was added to the sentence "The move from the coarse Zobler classes to the finer USDA classes is found to be more important to the mechanistic model **than to the empirical model**."

*P8, anoxic soil COS production: Did you consider to use the formulations of Meredith et al., 2018? These are similar to that of Whelan et al., 2016, but then with the alfa and beta parameters specific for peatland/wetland soils. That is, fca = 3700 for boreal peatland (Meredith et al., 2019) and alfa and beta for peatland from Meredith et al., (2018,). It would be interesting to compare those COS production estimates for wetland soils.*

Answer: In the mechanistic approach, the  $f_{CA}$  parameter is related to COS uptake by oxic soils only, as we consider anoxic soils as COS sources (equation 18). We did not use the formulation of Meredith et al. (2018) to represent soil COS flux for boreal peatlands as we decided to use the same formulation adapted from Ogée et al. (2016) for all anoxic soils. Indeed, soil COS emissions from peatlands are difficult to characterize as they can vary by an order of magnitude depending on the observation site (see Figure 1 in Meredith et al., 2018). This large variability shows that factors other than soil temperature could be important to include to represent and parametrize soil COS emissions from peatlands. Moreover, the wetland map from Tootchi et al. (2019) used in ORCHIDEE does not distinguish between the different wetland types. Therefore, improving soil COS flux modeling for the different types of anoxic soils should be the focus of future work as adapting the parameters to each wetland type might not be sufficient to improve their representation. However, as the approach to estimate soil COS fluxes from wetlands could be refined by distinguishing boreal peatland as suggested, we added this in section 4.3. as a future improvement. We completed the sentence as follows "We could also refine our approach by distinguishing between the different types of wetlands and define a  $P_{ref}$  value for each wetland type instead of a global value of 10 pmol COS  $m^{-2} s^{-1}$ . Then, a distinction could also be made for anoxic soil COS fluxes from boreal peatlands,

**as Meredith et al. (2019) give a value of $f_{CA}$ specific to this biome."**

*P10, L330: "....the same training method than the one used in Spielmann et al." should be "....the same training method* **as** *the one used in Spielmann et al."*

Answer: Thank you, this error was corrected.

*P10, L333: If I understand the description in Wehr et al. (2017) right, the soil COS fluxes at US-HA are not based on eddy-covariance fluxes. It can be better described as flux-profile measurements, connected to CO2 soil chamber measurements and profiles.*

Answer: The Referee is right, we clarified the description of US-HA measurement methods for soil COS fluxes "At US-HA, soil COS fluxes in 2012 and 2013 were not directly measured but derived from **flux-profile measurements**, **connected to CO2 soil chamber measurements and profiles.** A sub-canopy flux gradient approach was used to partition canopy uptake from soil COS fluxes. For more information on this approach and its limitations, see Wehr et al. (2017)."

*P10,L350-351:"* The stations located in the northern Hemisphere sample air masses coming from the entire northern hemisphere domain above 30 degrees." The stations cover mainly North-America and actually Eurasia is hardly covered, so I would not agree with this statement.

Answer: The stations located in the Northern hemisphere cover mainly North America, however they are sensitive to air masses coming from the entire Northern hemisphere domain above 30 degrees based on atmospheric modeling, as shown in Remaud et al. (2022; Figure 1 and part 2.2.1).

P11, L384: what does "d" stand for?

Answer: The abbreviation "80 or 667 d" was replaced by "80 or 667 days".

P12, L412: spell out "DA".

Answer: The abbreviation "DA" was replaced by "Data assimilation".

Results:

*P13,* section 3.1.1. I think the authors could put more emphasis on the potential role of nitrogen fertilization on soil fluxes. E.g. the results of IT-CRO, an agricultural site, could be emphasized in this context. Also the overestimated COS uptake at AT-NEU and ES-LMA could be discussed in light of nitrogen fertilization.

Answer: Thank you, we emphasized the importance of nitrogen inputs at AT-NEU, ES-LMA and IT-CRO in the description of figure 2. "Besides, AT-NEU and ES-LMA are managed grassland sites with nitrogen inputs. Then, soil COS production could also be enhanced by a high nitrogen content as suggested by several studies (Kaisermann et al., 2018; Kitz et al., 2020; Spielmann et al., 2020), which is not represented in our models. The mechanistic model is able to represent a net COS production at IT-CRO but overestimates it. This might highlight the importance of adapting the production parameters ( $\alpha$  and  $\beta$ ) in this model to adequately represent a net COS production. In this model, the net soil COS production is related to an increase in soil temperature. However, it is to be noted that IT-CRO is an agricultural site with nitrogen fertilization. Therefore, soil COS production in the observations could also be enhanced by nitrogen inputs."

*P13, L468 (Table 3): I would consider showing Table 3 as a figure. The same also for Table 4, which could even be combined with a Fig. from Table 3.*

Answer: We inserted the RMSD values in Table 3 and Table 4 in Figure 2 and Figure 3, respectively.

---

## Author Comment (AC2)

Biogeosciences Discuss., referee comment RC2
https://doi.org/10.5194/bg-2021-281-RC2, 2022

[Figure]

**Comment on bg-2021-281**

Anonymous Referee #2
* * *
Referee comment on "Global modelling of soil carbonyl sulfide exchange" by Camille Abadie et al., Biogeosciences Discuss., https://doi.org/10.5194/bg-2021-281-RC2, 2022
* * *
*Abadie et al. simulated soil COS fluxes on a global scale using a mechanistic soil model in a land surface model ORCHIDEE, and evaluated the simulation results, from both the mechanical model and an empirical model based on scaling soil respiration, against 7 sites in the Northern Hemisphere. Furthermore, an atmospheric transport model LMDZ was used to investigate the contribution of different soil flux products to the latitudinal gradient of atmospheric COS concentrations. Moreover, sensitivity analyses were performed to reveal the importance of various parameters, which is useful to understand the control mechanisms of the soil COS fluxes. Note that the mechanistic model has been previously developed and published. Nevertheless, implementing the existing model in ORCHIDEE to study the global soil COS fluxes is desired. This is a very nice model study. The paper is well structured and very well written and is certainly suitable for the journal of Biogeosciences.*

*As has been noticed by the authors, the available field observations of soil COS fluxes are very limited, which is especially true when global COS fluxes are the focus of the study. In fact, all 7 sites are located in a narrow latitude range of 42 – 62 °N, and do not cover a full seasonal cycle, which makes it difficult to evaluate the simulation results on a global scale, and raises many questions around whether the presented simulation results are justified, e.g., whether smaller global COS soil flux than previous estimates is trustworthy, whether very large emissions in part of the tropics exist, not to mention the validation of the seasonal cycle and the diel cycle of the simulated results. On the other hand, these are also nice topics to be followed on. For this manuscript, I strongly feel that these points need to be better clarified in the revised version before publication.*

Answer: The authors thank the Referee for the generally positive comment and for the very insightful questions and suggestions to improve this manuscript.

We agree with the Referee, the scarcity of soil COS flux observations and the limited latitudinal range are important limitations to the validation of this study. However, we selected observation sites that cover the largest diversity given the available observations, which represent 7 different plant functional types in ORCHIDEE. Moreover, previous studies had access to an even smaller number of observations to validate their estimates as many field observations of soil COS fluxes were carried in the recent years. The difficulty of performing long-term and continuous measurements of soil COS fluxes also has to be acknowledge as flux towers do not allow to measure soil COS flux only. The efforts made to collect soil COS flux observations enabled the recent study of Kooijmans et al. (2021), and this study to benefit from the increasing number of measurements. As indicated by the Referee, more observations in the tropics would help to validate the net soil COS production simulated in some regions, such as the large one found in Northern India in both SiB4 and

ORCHIDEE. Soil chamber measurements of soil COS flux were performed at La Selva Biological Station, at a tropical rainforest in Costa Rica (Sun et al., 2014). When available, these observations could allow a first comparison with the simulated soil COS flux in the tropics. We completed section 4.3. as follows "Moreover, one difficulty with the study of soil COS fluxes arises from the scarcity of field measurements that could be used for model validation and calibration. **Besides, the observation sites considered here are all located in a small latitudinal range between 39°N and 62°N. Measurements in the tropics and in the Southern hemisphere are needed. Especially, soil COS flux observations in Northern India could help to validate the net soil COS production simulated in both SiB4 and ORCHIDEE. In the tropical rainforest, soil COS flux measurements were performed at La Selva Biological Station in Costa Rica (Sun et al., 2014). When available, these measurements could allow a first comparison between the observed and simulated soil COS flux in a tropical region.**"

Then, a comparison to another global model can also give some strength to our results. The study of the mechanistic soil COS model presented in Kooijmans et al. (2021) enables us to compare our results to those obtained in SiB4. We emphasized this comparison in the Results part as the distribution between net soil COS sources and net soil COS sinks at the global scale shows some similarities between SiB4 and ORCHIDEE. This was added to section 3.2.1. "**The distribution and magnitude of soil COS flux from the empirical approach is similar to the one presented in Kooijmans et al. (2021) (see Figure S15 in the supplementary material of Kooijmans et al., 2021), when implemented in SiB4. For the mechanistic model, the comparison of oxic soil COS flux distribution with the one in SiB4 shows a net soil COS emission in India in both SiB4 and ORCHIDEE. However, the maximum oxic soil COS flux is about 60 pmol m$^{-2}$ s$^{-1}$ higher in ORCHIDEE than in SiB4. The regions with the strongest net oxic soil COS uptake also differ between SiB4 and ORCHIDEE as it is concentrated in the tropics in SiB4 and in Western North and South America, and in China for ORCHIDEE.**"

We then added the net global budget for oxic soils computed in SiB4 in the Discussion part in section 4.1., which also concludes to a smaller contribution from oxic soils than estimated in previous studies. The discussion was completed as follows "According to the mechanistic approach of this study, the COS budget for oxic soil is a net sink of -126 GgS yr$^{-1}$ over 2009-2016, which is close to the value of -130 GgS yr$^{-1}$ found by Kettle et al. (2002) (**Erreur ! Source du renvoi introuvable.**3). **This net COS uptake by oxic soils is higher than the one found in SiB4 by Kooijmans et al. (2021) with -89 GgS yr$^{-1}$, also based on the mechanistic model described in Ogée et al. (2016). In SiB4 and in ORCHIDEE**, the mechanistic model gives the lowest oxic soil COS net uptake compared to all previous studies, which were all using empirical approaches."

Sun, W., Maseyk, K. S., Juarez, S., Lett, C., and Seibt, U. H.: Soil-atmosphere carbonyl sulfide (COS) exchange in a tropical rainforest at La Selva, Costa Rica. AGU Fall Meeting Abstracts, 2014, B41C-0075, 2014.

*Regarding the selection of the field sites, why were the soil flux measurements in an agricultural field in the Southern Great Plains by Maseyk et al., 2014 not used?*

Answer: The data from Maseyk et al. (2014) are not publicly available and we were unable to obtain the original data. The agricultural field in the Southern Great Plains would be represented by C3 crops in ORCHIDEE. In the dataset we used, IT-CRO site is also represented by C3 crops.

*L25: remove "budgets" after "atmospheric COS"*

Answer: "Budgets" was removed as suggested.

*L37 specify the region of the tropics, otherwise, it sounds like high emissions in all tropical regions*

Answer: The Referee is right, we have now replaced the sentence "The predicted spatial

distribution of soil COS fluxes, with large emissions in the tropics from oxic (up to 68.2 pmol COS m-2 s-1) and anoxic (up to 36.8 pmol COS m-2 s-1) soils in the tropics mainly, marginally improves the latitudinal gradient of atmospheric COS concentrations, after transport by the LMDZ atmospheric transport model" by "The predicted spatial distribution of soil COS fluxes, with large emissions from oxic (up to 68.2 pmol COS m-2 s-1) and anoxic (up to 36.8 pmol COS m-2 s-1) soils **in the tropics, especially in India and in the Sahel region**, marginally improves the latitudinal gradient of atmospheric COS concentrations, after transport by the LMDZ atmospheric transport model."

*L170: please briefly discuss why the steady-state condition is valid? what assumption has to be made to make the steady-state valid?*

Answer: We assume that, over the 30-minute time step used here to run the ORCHIDEE model, the environmental conditions in the soil (temperature, moisture, etc.) are constant. We also assume chemical equilibrium between the gaseous and dissolved COS and neglect COS adsorption on soil particles, as suggested by Ogée et al. (2016). In these conditions, gas exchange of COS in the soil column is evolving rapidly to a steady state, attained before the end of the 30-minute time step. For example, if we consider COS diffusion, the diffusion time scale is $L^2/D$ with L the diffusion length and D the diffusivity of COS. In ORCHIDEE, we consider that COS uptake happens in the top 9 cm. With the diffusivity of COS in the air that is around $1.4*10^{-5}$ m²/s, the diffusion time scale is around 10 minutes. Moreover, the first order reaction from the carbonic anhydrase enzyme COS consumption reduces the time needed to reach the steady-state condition. The evolution of the soil COS flux can therefore be approximated as a succession of steady states from one time step to the other. The model description was completed as follows "Under steady-state conditions and uniform soil temperature, moisture and porosity profiles, an analytical solution of Eq. 2 can be found (Ogée et al., 2016). **We assume that the environmental conditions, such as soil temperature and moisture, are constant in ORCHIDEE over the 30-minute model time step. We also assume chemical equilibrium between the gaseous and the dissolved COS, neglecting advection as suggested by Ogée et al. (2016). In these conditions, the typical time scale for COS diffusion in the upper active soil layer is much shorter than the 30-minute model time step."**

*L457-458: Clearly, the mechanistic model predicts nearly no seasonal cycle except for large production signals in the summertime, as is shown in Figure 2. However, this implies that relatively large net soil uptake exists in northern high latitude in winter times, when the temperature can be rather low and the land is covered by snow. This makes me wonder what the applicable range for the parameters shown in the method section, e.g., the valid temperature range of $f_{CA}$ in eq. 16, the valid temperature range for α and β in eq. 17.*

Answer: We acknowledge that the impact of snow cover on soil COS flux is not represented in the soil COS models and it could be important in specific cases. However, the scarcity of COS flux observations from soils covered by snow and in winter does not allow to implement its effect. Note that Helmig et al. (2009) found that COS uptake was not zero when soil is covered by snow at Niwot Ridge, Colorado. We added in section 4.3. "**In the soil COS models, the impact of snow cover is also not represented. Indeed, due to the scarcity of soil COS flux observations in winter and with snow cover, its effect on soil COS flux could not be implemented in soil COS models yet**. **However, Helmig et al. (2009) found that COS uptake was not zero when soil is covered by snow at Niwot Ridge, Colorado."**
As mentioned by the Referee, at the sites where there is no strong soil COS production for the mechanistic model, the absence of seasonality leads to a winter soil COS flux that is of the same order of magnitude as the summertime flux. However, for the simulated flux it is to be noted that despite a similar order of magnitude, the soil COS flux in summer does not result only from soil COS uptake but from the combination of soil COS uptake and production, which compensates for the increase in soil COS uptake in the summertime (Figure C1). Observations of soil COS fluxes during wintertime would be of much help to

evaluate this net soil COS uptake simulated in winter.

Concerning the $f_{CA}$ parameter for soil COS uptake, in Ogée et al. (2016) soil COS fluxes were simulated with several $f_{CA}$ values for temperatures ranging from 0°C to 25°C. Then, for the production term with the $\alpha$ and $\beta$ parameters, the temperature range tested in Whelan et al. (2016) was set between 10°C and 40°C. At the studied sites, the temperature can reach lower values than 10°C. However, in the absence of more information on soil COS production in winter, we kept this expression of soil COS production for lower temperatures which implies that the production term would tend to zero.

Helmig, D., Apel, E., Blake, D. *et al.* Release and uptake of volatile inorganic and organic gases through the snowpack at Niwot Ridge, Colorado. *Biogeochemistry* **95,** 167–183 (2009). https://doi.org/10.1007/s10533-009-9326-8

*L471-473: Note that vegetation was also removed for the FI-HYY site, as is in Sun et al., 2018 "The moss layer or any other vegetation was removed to expose the humus layer inside the chambers." This contradicts the statement. If the assumption would be true, what would be the mechanism for artificially enhanced COS production?*

Answer: The statement was "However, the mechanistic model struggles to reproduce soil COS fluxes at AT-NEU and ES-LMA, with an overestimation of soil COS uptake or an underestimation of soil COS production at AT-NEU and a delay in the simulated net COS production at ES-LMA. We might suspect that the removal of vegetation at these sites prior to the measurements could have artificially enhanced COS production in the observations". This assumption was based on the fact that removing vegetation can affect soil structure and increase the accessibility of plant matter residues and more generally soil organic matter to degradation and abiotic COS production. The enhancement of soil COS production by a degradation of accessible soil organic matter was discussed in Whelan et al. (2016).

Indeed, vegetation was also removed at FI-HYY. We added this sentence in section 2.2.2. concerning the soil COS flux measurements at FI-HYY "**Any vegetation was removed from the chambers before the measurements**."

However, we think that the removal of vegetation at FI-HYY might not have the same impact on soil COS production as at AT-NEU and ES-LMA as these sites have different environmental conditions. At AT-NEU and ES-LMA, the average soil temperature (respectively 19.8°C and 20.5°C with maximums of 34.7°C and 28.3°C) during the observation periods is higher than at FI-HYY (10.9°C with a maximum of 14.9°C). At these two sites, soil is also expected to receive more direct radiations than Hyytiälä forest, as they are grasslands. Soil COS production was related to thermal or photo-degradation of organic matter (Kitz et al., 2017, 2020; Whelan et Rhew, 2015; Whelan et al., 2016, 2018). Therefore, the authors would disagree with the fact that the absence of net soil COS production despite the removal of vegetation at FI-HYY contradicts a possible enhancement of soil COS production with the removal of vegetation at AT-NEU or ES-LMA. We added this justification "We might suspect that the removal of vegetation at these sites prior to the measurements could have artificially enhanced COS production in the observations. **Indeed, the removal of vegetation could change soil structure and increase the availability of vegetation residues and soil organic matter to degradation (Whelan et al., 2016). AT-NEU and ES-LMA are grassland sites for which soils are expected to receive higher light intensity than forest soils. These sites also show a high mean soil temperature of about 20°C during the measurement periods. Therefore, high soil temperature and light intensity on soil surface could enhanced soil COS production as it was related to thermal or photo degradation of soil organic matter (Kitz et al., 2017, 2020; Whelan et Rhew, 2015; Whelan et al., 2016, 2018). This is not the case at FI-HYY, ET-JA or DK-SOR, where soil temperature is much lower (mean value about 10°C at FI-HYY and 15°C at ET-JA and DK-SOR during the measurement periods) and the forested cover decreases the radiation level reaching the soil. Note that herbaceous biomass is also likely to be higher in grasslands than in forests.**"

*L478: The diel cycles of simulated COS soil fluxes by the mechanistic model shown in Figure 3 are totally not supported by the observations. Note that when relatively large uncertainties in the observations are considered, a minimum net soil COS uptake in the*

*observations is not significant at all. Actually, the large discrepancy calls for a better understanding of the mechanistic model: what causes the diel cycles in the model but not shown in the observations.*

Answer: We agree and revised our analysis for the diel cycle at US-HA that shows large variability. We removed the sentence "A minimum net soil COS uptake is also observed at US-HA but in the afternoon". As suggested by the Referee, we studied the response of soil COS flux diel cycle to soil temperature and soil water content to better characterize the difference between the simulated and observed diel cycle.

Simulated soil COS flux:
Figure 3 shows a diel cycle of soil COS flux simulated with the mechanistic model at IT-CRO and US-HA. We represented below the modelled mechanistic soil COS flux versus soil temperature and soil water content in a similar way as in Figure 4 but with the hourly average fluxes of the mean diel cycle represented in Figure 3. As shown in the figure below, the main driver of the diel cycle at US-HA and IT-CRO for the mechanistic model is soil temperature, leading to a decrease of the net soil COS uptake at US-HA and to an increase of the net soil COS production at IT-CRO. The simulated ranges of soil temperature at US-HA and IT-CRO correspond to the highest values compared to the other sites. It is also to be noted that the simulated soil water content at all sites shows almost no diel variations (about 1%).

[Figure]

**Figure RC2_1: Simulated hourly average net soil COS flux (pmol m² s⁻¹) versus soil temperature (°C) and soil water content (SWC) (m³.m⁻³) at AT-NEU, ES-LMA, IT-CRO, DK-SOR, ET-JA, US-HA and FI-HYY, for the mechanistic model. Soil COS fluxes represent the mean diel cycle of net soil COS fluxes (pmol m⁻² s⁻¹) over a month at: AT-NEU (08/2015), ES-LMA (05/2016), IT-CRO (07/2017), DK-SOR (06/2016), ET-JA (08/2016), FI-HYY (08/2015) and US-HA (07/2012).**

Observed soil COS flux:
We also represented below the hourly average of the observed soil COS flux used to compute the mean diel cycle in Figure 3 versus soil temperature and soil water content. The observed soil COS fluxes show diel variations at AT-NEU, ES-LMA, IT-CRO and DK-SOR illustrated in Figure 3. At these four sites, soil COS flux exhibits an increasing trend of the net soil COS production with soil temperature at AT-NEU, ES-LMA and IT-CRO, and a decreasing trend of the net soil COS uptake with soil temperature at DK-SOR. However, Spielmann et al. (2020) found that the random forest regression of soil COS fluxes was the

most sensitive to the incident shortwave radiation, which is not represented in the soil COS models. Kitz et al. (2020) also related daytime net COS production to high light intensity reaching the soil surface at ES-LMA.

At US-HA, no diel cycle is found in the observations (Figure 3) while the mechanistic model shows a decrease in the net soil COS uptake around 3pm. As seen above, the mechanistic model diel cycle at US-HA is a response of soil temperature. However at this site, Wehr et al. (2017) indicate that they did not find a dependency of soil COS flux to soil temperature or soil water content as their ranges of variation could be too low to see an effect. Indeed, the simulated range of soil temperature at US-HA in ORCHIDEE is larger than the one measured on site. The diel variations of soil COS flux simulated at US-HA could be due to this larger range of soil temperature enhancing soil COS production, while this effect is not found in the observations, or more compensated by soil COS uptake than in the simulated flux.

[Figure]

**Figure RC2_2: Observed hourly average net soil COS flux (pmol m² s⁻¹) versus soil temperature (°C) and soil water content (SWC) (m³.m⁻³) at AT-NEU, ES-LMA, IT-CRO, DK-SOR, ET-JA, US-HA and FI-HYY. Soil COS fluxes represent the mean diel cycle of net soil COS fluxes (pmol m⁻² s⁻¹) over a month at: AT-NEU (08/2015), ES-LMA (05/2016), IT-CRO (07/2017), DK-SOR (06/2016), ET-JA (08/2016), FI-HYY (08/2015) and US-HA (07/2012).**

The part on soil COS flux simulated with the mechanistic model in section 3.1.2. was modified as follows "Figure 3 shows the comparison between the simulated and observed mean diel cycles over a month. The observations show a minimum net soil COS uptake or a maximum net soil COS production reached between 11 am and 1 pm at AT-NEU, ES-LMA, IT-CRO and DK-SOR. At AT-NEU and ES-LMA, neither model is able to represent the observed diel cycle. **At these grassland sites, Spielmann et al. (2020) and Kitz et al. (2020) found that the daytime net COS emissions were mainly related to high radiations reaching the soil surface, which impact is not represented in the soil COS models**. At IT-CRO and DK-SOR, the diel cycles simulated by the mechanistic model show patterns similar to the observations with a peak in the middle of the day, but with an overestimation of the net soil COS production and a delay in the peak at IT-CRO, and an overestimation of the net soil COS uptake at DK-SOR. The mechanistic model reproduces the absence of a diel cycle observed at FI-HYY and ET-JA but with an underestimation of the net soil COS uptake at ET-JA. **AT US-HA, the observed soil COS flux does not exhibit diel variations while the mechanistic model shows a peak with a decrease of the net soil COS uptake around 3 pm. Wehr et al. (2017) explain this absence of diel**

**cycle in the observations by a range of variations for soil temperature and soil water content that is too low to influence soil COS flux. In ORCHIDEE, the simulated range of temperature at US-HA is larger than the one measured on site and temperature is the main driver of the decrease in net soil COS uptake at this site (not shown). Therefore, the enhancement of soil COS production by soil temperature could be only found in the simulated flux. Another possibility is that it could be totally compensated by soil COS uptake in the observations**."

*L568: Section 3.1.5, although it is nice that the authors have made an effort to optimize soil COS flux, it may be premature. As the results are not used in the following results, I suggest leaving this section out or putting it into the Appendix.*

Answer: Indeed, the optimized parameters are not used at the global scale as there are only few sites with enough observations to perform an optimization. However, the optimization gives useful information as it shows that for the empirical model, improving the representation of soil COS flux leads to a degradation of the simulated water content at FI-HYY, while soil water content is improved after soil COS flux assimilation for the mechanistic model. We added the RMSD change of soil water content prior and post optimization in section 3.1.5. "However, while it improves the simulated water content compared to the observations for the mechanistic model at the two sites **(RMSD decreases by 28% at FI-HYY and 22% at US-HA)**, it leads to a degradation at FI-HYY for the empirical model **(RMSD increases by more than 3 times)**". The optimization also shows how the most important parameters of the mechanistic model are affected when assimilating observed soil COS fluxes. As the optimization does not only aim at improving the simulated soil COS flux, but also at better understanding the models and their limitations, the authors would like to keep this study as part of the manuscript.

*L645: Section 3.2.2 Temporal evolution of the soil COS budget. It is expected that oxic soil COS sinks would decrease when atmospheric COS concentration decreases, and one could even expect that the decrease is, to the first order, proportional to the decrease of COS concentrations. However, the sharp decrease from 2016 is far beyond this. What are then the main reasons that can explain the sharp decrease in the mechanistic model?*

Answer: Indeed, the decrease in oxic soil COS budget computed with the mechanistic model is sharper than the drop in atmospheric COS concentration. Atmospheric COS concentration is not the only driver of soil COS flux in the mechanistic model. Changes in the soil COS production term also impact COS flux from oxic soils, which depends on soil temperature. Between 2010 and 2019, the soil COS production term has increased by 7%. In this model, the production term can have a strong impact on the net soil COS flux as it is expressed as an exponential response to soil temperature. In addition to a decrease of soil COS diffusion into the soil with lower atmospheric COS concentrations, an increase in the production term contributes to reducing the net soil COS uptake by oxic soils.

It is also to be noted that the drop in atmospheric COS concentration is not homogenous around the globe as illustrated in the figure below which shows the difference in simulated atmospheric COS concentrations between 2010 and 2019. The largest simulated decreases are located in Europe and South of China. Then, in these regions, the soil COS uptake is expected to be particularly affected by the drop in atmospheric COS concentration.

[Figure]

**Figure RC2_3: Difference of simulated atmospheric COS concentrations (ppt) between 2010 and 2019. Negative values show a loss in atmospheric COS concentration in 2019 compared to 2010.**

The simulated changes in soil temperature and moisture are also heterogenous as illustrated by the difference of soil temperatures and moisture between 2010 and 2019 (positive values represent an increase in 2019 compared to 2010). Changes in oxic soil COS budget result from the combined effect of decreasing atmospheric COS concentration and changes in the drivers of soil COS fluxes.

[Figure]

**Figure RC2_4: Difference of simulated soil temperaturas (°C) between 2010 and 2019. Positive values show an increase in soil temperature in 2019 compared to 2010.**

[Figure]

**Figure RC2_5: Difference of simulated soil water contents (%) between 2010 and 2019. Negative values show a decrease in soil water content in 2019 compared to 2010.**

Section 3.2.2 was completed as follows "**Note that the decrease in oxic soil COS budget computed with the mechanistic model is sharper than the drop in atmospheric COS concentration because changes in oxic soil COS budget result from the combined effect of decreasing atmospheric COS concentration and changes in the drivers of soil COS fluxes (i.e., changes in soil temperature and water content during the 10 year period which are not homogenously distributed around the globe (not shown)).**"